# Contribution of dorsal horn CGRP-expressing interneurons to mechanical sensitivity

Line S Löken[1†*], Joao M Braz[1], Alexander Etlin[1], Mahsa Sadeghi[1], Mollie Bernstein[1], Madison Jewell[1], Marilyn Steyert[1], Julia Kuhn[1], Katherine Hamel[1], Ida J Llewellyn-Smith[2,3], Allan Basbaum[1*]

[1]Department Anatomy University California, San Francisco, San Francisco, United States; [2]Discipline of Physiology, Adelaide Medical School, University of Adelaide, Adelaide, Australia; [3]Department of Cardiology, Flinders Medical Centre, Bedford Park, Australia

**Abstract** Primary sensory neurons are generally considered the only source of dorsal horn calcitonin gene-related peptide (CGRP), a neuropeptide critical to the transmission of pain messages. Using a tamoxifen-inducible $Calca^{CreER}$ transgenic mouse, here we identified a distinct population of CGRP-expressing excitatory interneurons in lamina III of the spinal cord dorsal horn and trigeminal nucleus caudalis. These interneurons have spine-laden, dorsally directed, dendrites, and ventrally directed axons. As under resting conditions, CGRP interneurons are under tonic inhibitory control, neither innocuous nor noxious stimulation provoked significant Fos expression in these neurons. However, synchronous, electrical non-nociceptive Aβ primary afferent stimulation of dorsal roots depolarized the CGRP interneurons, consistent with their receipt of a VGLUT1 innervation. On the other hand, chemogenetic activation of the neurons produced a mechanical hypersensitivity in response to von Frey stimulation, whereas their caspase-mediated ablation led to mechanical hyposensitivity. Finally, after partial peripheral nerve injury, innocuous stimulation (brush) induced significant Fos expression in the CGRP interneurons. These findings suggest that CGRP interneurons become hyperexcitable and contribute either to ascending circuits originating in deep dorsal horn or to the reflex circuits in baseline conditions, but not in the setting of nerve injury.

*For correspondence:
line.loken@gu.se (LSL);
allan.basbaum@ucsf.edu (AB)

†Present affiliation: Institute of Neuroscience and Physiology, Department of Physiology, University of Gothenburg, Gothenburg, Sweden.

## Introduction

Calcitonin gene-related peptide (CGRP) is the most prominent molecular marker of the peptidergic subpopulation of primary afferent nociceptors (*Basbaum et al., 2009*). When released from peripheral terminals of sensory neurons, CGRP acts on endothelial cells that line blood vessels, producing pronounced vasodilation (*Brain et al., 1985*). Recent efforts to develop novel therapeutics in the management of migraine led to the successful development of antibodies that scavenge CGRP, reducing the vasodilation that triggers migraine (*Ho et al., 2010*). When released into the superficial dorsal horn from the central branches of sensory neurons, CGRP, along with its co-occurring neuropeptide, substance P, potentiates the glutamatergic excitation of postsynaptic neurons, contributing to injury-provoked central sensitization (*Ryu et al., 1988*; *Woolf and Wiesenfeld-Hallin, 1986*). The latter process, in turn, contributes to the ongoing pain and profound hypersensitivity characteristic of both inflammatory and neuropathic pains. Interestingly, a recent study showed that pharmacological inhibition of CGRP receptor signaling in the periphery alleviates incision-induced mechanical and heat hypersensitivity, but not neuropathic pain, suggesting that primary sensory neuron-derived CGRP differentially influences injury-induced persistent pain (*Cowie et al., 2018*).

**eLife digest** The ability to sense pain is critical to our survival. Normally, pain is provoked by intense heat or cold temperatures, strong force or a chemical stimulus, for example, capsaicin, the pain-provoking substance in chili peppers. However, if nerve fibers in the arms or legs are damaged, pain can occur in response to touch or pressure stimuli that are normally painless. This hypersensitivity is called mechanical allodynia.

A protein called calcitonin gene-related peptide, or CGRP, has been implicated in mechanical allodynia and other chronic pain conditions, such as migraine. CGRP is found in, and released from, the neurons that receive and transmit pain messages from tissues, such as skin and muscles, to the spinal cord. However, only a few distinct groups of CGRP-expressing neurons have been identified and it is unclear if these nerve cells also contribute to mechanical allodynia.

To investigate this, Löken et al. genetically engineered mice so that all nerve cells containing CGRP produced red fluorescent light when illuminated with a laser. This included a previously unexplored group of CGRP-expressing neurons found in a part of the spinal cord that is known to receive information about non-painful stimuli. Using neuroanatomical methods, Löken et al. monitored the activity of these neurons in response to various stimuli, before and after a partial nerve injury. This partial injury was induced via a surgery that cut off a few, but not all, branches of a key leg nerve.

The experiments showed that in their normal state, the CGRP-expressing neurons hardly responded to mechanical stimulation. In fact, it was difficult to establish what they normally respond to. However, after a nerve injury, brushing the mice's skin evoked significant activity in these cells. Moreover, when these CGRP cells were artificially stimulated, the stimulation induced hypersensitivity to mechanical stimuli, even when the mice had no nerve damage. These results suggest that this group of neurons, which are normally suppressed, can become hyperexcitable and contribute to the development of mechanical allodynia.

In summary, Löken et al. have identified a group of nerve cells in the spinal cord that process mechanical information and contribute to touch-evoked pain. Future studies will identify the nerve circuits that are targeted by CGRP released from these nerve cells. These circuits represent a new therapeutic target for managing chronic pain conditions related to nerve damage, specifically mechanical allodynia, which is the most common complaint of patients with chronic pain.

Despite much earlier reports, which used colchicine to enhance somatic CGRP levels (*Kruger et al., 1988*; *Tie-Jun et al., 2001*) and a more recent report (*McCoy et al., 2012*) of small CGRP-positive cells in the dorsal horn of a reporter mouse, the prevailing view is that dorsal horn CGRP derives exclusively from afferents. Here, we took advantage of a tamoxifen-inducible $Calca^{CreER}$ mouse line, which when crossed with a tdTomato reporter mouse, reveals a discrete population of CGRP-expressing interneurons that are concentrated in lamina III and inner lamina II of the spinal cord dorsal horn and trigeminal nucleus caudalis. Unlike dorsal horn vertical cells, which have ventrally directed dendrites and a dorsally directed axon, the CGRP interneurons have mainly dorsally directed dendrites and ventrally directed axons. A comprehensive functional analysis showed that these interneurons are minimally responsive to a host of acute, innocuous or noxious mechanical and chemical stimuli, despite the fact that electrical stimulation of Aβ afferents readily activates the cells. On the other hand, an innocuous mechanical stimulus evoked significant Fos expression in the setting of peripheral nerve injury and chemogenetic activation of the interneurons produced clear mechanical hypersensitivity. Conversely, caspase-mediated ablation of the neurons increased mechanical thresholds. We conclude that these CGRP-expressing interneurons engage deep dorsal horn nociresponsive circuits that contribute either to ascending circuits originating in deep dorsal horn or to the reflex circuits in baseline conditions, but not in the setting of nerve injury.

## Results

To map the distribution of CGRP-expressing neurons in the dorsal horn, we first crossed the $Calca^{CreER}$ mouse line with a $ROSA26^{fs-tdTomato}$ (Ai14) mouse line, hereafter referred to as CGRP-tdTomato. Adult mice were administered tamoxifen twice (150 mg/kg, at postnatal days 21–23), and as

reported previously, this triggered tdTomato expression in primary sensory neurons (*Patil et al., 2018*). However, we also recorded significant labeling of neurons in the dorsal horn and trigeminal nucleus caudalis (N. Caudalis; *Figure 1*). Importantly, because the tamoxifen is administered at 3–4 weeks of age, we conclude that the pattern of expression is reflective of that found in the adult.

We first confirmed the approach by ensuring that the tdTomato-expressing primary sensory neurons of the dorsal root ganglia (DRG) double-label with an antibody to CGRP. *Figure 1* illustrates that 80% of tdTomato-positive neurons in trigeminal ganglia (TG) and DRG immunostained for CGRP and that 78% of the CGRP immunoreactive neurons were tdTomato-positive (*Figure 1a–d*).

Consistent with the central projection of CGRP-expressing sensory neurons, we also observed very dense tdTomato-positive terminals in the superficial laminae of the dorsal horn and nucleus caudalis. We also recorded many tdTomato-labeled neurons in regions of the central nervous system known to contain significant populations of CGRP-immunoreactive neurons or terminals, including motoneurons in the ventral horn of the spinal cord (*Figure 1—figure supplement 1*), the parabrachial nucleus (*Figure 1—figure supplement 2*), subparafascicularis of the thalamus (*Figure 1—figure supplement 3*; *Yasui et al., 1991*), and central nucleus of the amygdala (*Figure 1—figure supplement 3*) and in cranial motor nuclei (*Figure 1—figure supplement 4*). We conclude that the pattern of CGRP-expression observed in the *Calca^{CreER}* mouse provides a reliable marker of CGRP-expressing neurons in the adult.

Unexpectedly, we also found large numbers of small tdTomato-positive neurons in the superficial dorsal horn and nucleus caudalis (notably in lamina III) and occasionally in more superficial layers (*Figure 1e–f*: *Figure 8—figure supplement 1c*). Consistent with previous literature, we did not detect CGRP-immunoreactivity in dorsal horn neurons using well-validated antibodies. However, by in situ hybridization we confirmed that *Calca* mRNA is present in neurons in the same regions of the spinal cord dorsal horn and nucleus caudalis (*Figure 1g–h*), which is consistent with the single cell PCR reports of *Calca* message in subpopulations of dorsal horn neurons (*Häring et al., 2018*; *Sathyamurthy et al., 2018*). We speculate that the lack of CGRP immunostaining reflects rapid transport of the peptide from the cell body to its axon, which undoubtedly underlies the requirement for colchicine to demonstrate these neurons by immunocytochemistry (*Kruger et al., 1988*; *Tie-Jun et al., 2001*). We found the CGRP-positive interneurons to be particularly abundant at the most caudal levels of the nucleus caudalis, markedly decreasing rostrally as the hypoglossal nucleus appears (*Figure 1—figure supplement 4*).

## CGRP dorsal horn neurons are excitatory interneurons

We next asked whether these CGRP-expressing neurons include both projection and interneurons. First, we injected the retrograde tracer Fluorogold (1%) into several brain areas that receive projections from the spinal cord dorsal horn. Despite an extensive analysis, which included injections into the ventrobasal and nucleus submedius (*Yoshida et al., 1991*) of the thalamus, lateral parabrachial nucleus (see *Figure 5—figure supplement 2b–c* for injection site in the parabrachial nucleus), and dorsal column nuclei, which are targeted by postsynaptic dorsal column neurons located in the region of lamina IV of the dorsal horn, we found no evidence of CGRP-expressing projection neurons. This finding was confirmed with an anterograde-tracing approach in which we injected an AAV1-flex-GCaMP6s virus unilaterally into the nucleus caudalis of *Calca^{CreER}*/tdTomato mice (*Figure 5—figure supplement 2a*). After 4 weeks, we examined the brainstem, thalamus, and hypothalamus for GFP-labeled fibers, but found no evidence of long-distance axonal projections deriving from the lamina III CGRP cells.

By immunolabeling the CGRP-tdTomato neurons, we next determined that these cells are excitatory and define a unique subset of interneurons. First, the CGRP-tdTomato cells co-express Lmx1b (98%; 92/94 tdTomato cells), but not Pax2 (*Figure 2—figure supplement 1*), which are excitatory and inhibitory markers, respectively. Some of the CGRP-tdTomato cells populate inner lamina II, and here approximately 16% co-expressed PKCγ (31/187 tdTomato cells), a marker of a large population of excitatory interneurons (*Malmberg et al., 1997*). Sixty-three (97/158 tdTomato cells) and 9% (9/97 tdTomato cells) of the CGRP interneurons co-expressed calbindin and calretinin, respectively, calcium binding proteins that mark subpopulations of excitatory dorsal horn interneurons (*Figure 2*). The incomplete immunohistochemical overlap with major neurochemical classes of dorsal horn interneurons indicates that the CGRP interneurons are heterogeneous consistent with previously described populations of dorsal horn neurons. However, as there is a limited number of quality

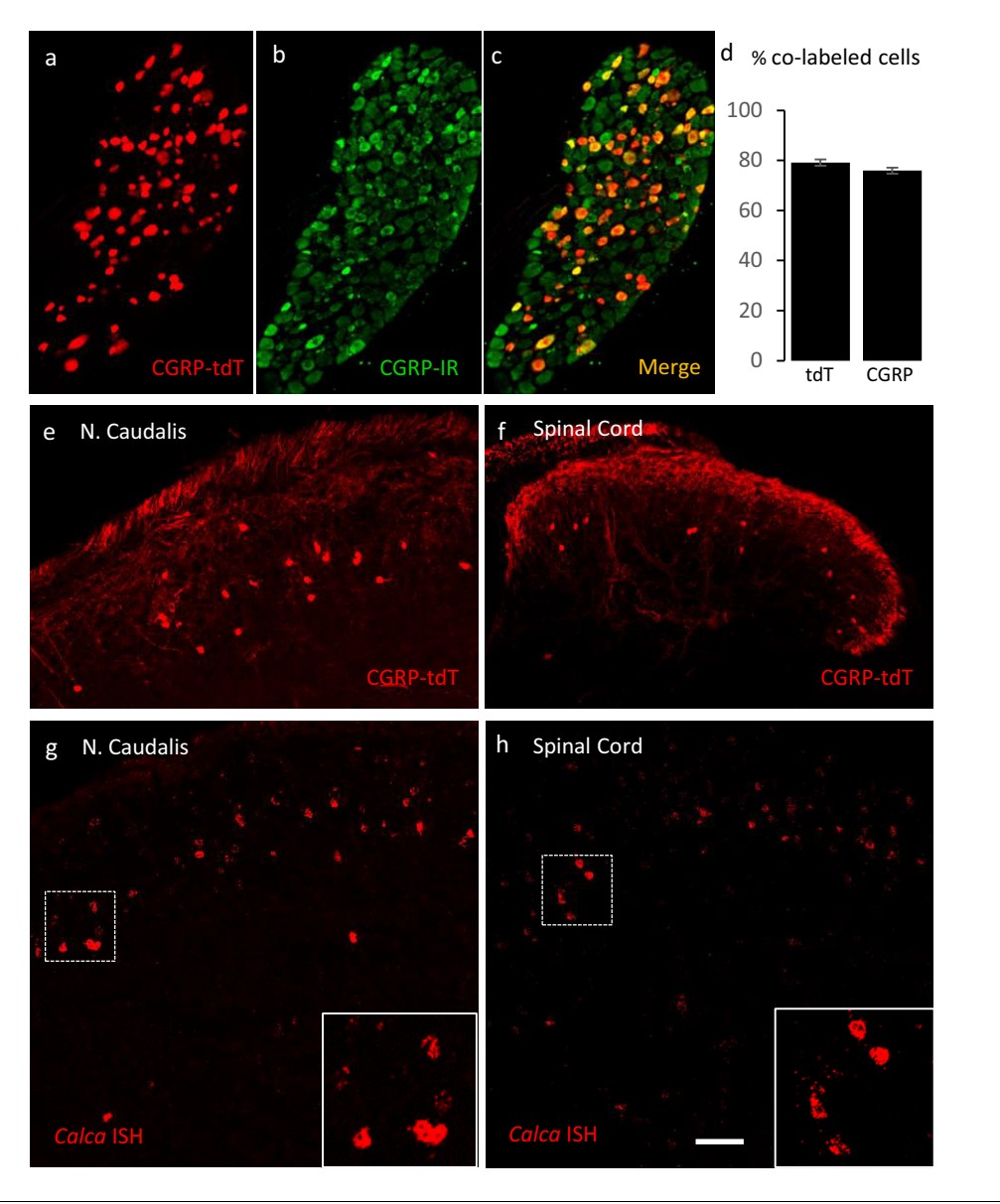

**Figure 1.** Validating the *Calca^CreER* transgenic mouse. (a–c) Example of genetically labeled CGRP neurons from dorsal root ganglion of double transgenic *Calca^CreER*/tdTomato mice generated by crossing the *Calca^CreER* mouse line with a *ROSA26^fs-tdTomato* (Ai14) mouse line (CGRP-tdTomato). Adult Calca^CreER/tdTomato mice received two injections of tamoxifen (150 mg/kg). Co-localization of tdTomato-(red) with CGRP- immunoreactivity (green) confirmed the specificity of *Calca^CreER* expression in trigeminal and dorsal root ganglia. (d) 80% of tdTomato-positive neurons were immunoreactive for CGRP (left bar) and 78% of CGRP-positive neurons were tdTomato-immunoreactive (right bar). Bars show mean and standard error (SEM) (three mice, four sections each). (e–f) CGRP-tdTomato expression was also detected in neurons of nucleus caudalis (e) and the spinal cord dorsal horn (f). The CGRP-tdTomato-immunoreactive neurons were concentrated in lamina III and occasionally observed in more superficial layers. The CGRP-tdTomato-labeled neurons were also abundant in regions of the central nervous system known to contain significant populations of CGRP-immunoreactive neurons or terminals (*Figure 1—figure supplements 1–4*). (g–h) In situ hybridization confirmed expression of *Calca* mRNA in the dorsal horn (g) and nucleus caudalis (h). Insets show higher magnification of the *Calca* mRNA expressing neurons. Scale bars: 100 µm. The online version of this article includes the following figure supplement(s) for figure 1:

**Figure supplement 1.** CGRP-tdTomato expression in the lumbar spinal cord.
**Figure supplement 2.** CGRP-tdTomato expression in the parabrachial nucleus.
**Figure supplement 3.** CGRP-tdTomato expression in the amygdala.

*Figure 1 continued on next page*

*Figure 1 continued*

**Figure supplement 4.** CGRP-tdTomato expression in the trigeminal nucleus caudalis.

antibodies that can be used for comprehensive neurochemical profiling we turned to in situ hybridization (*Figure 3*).

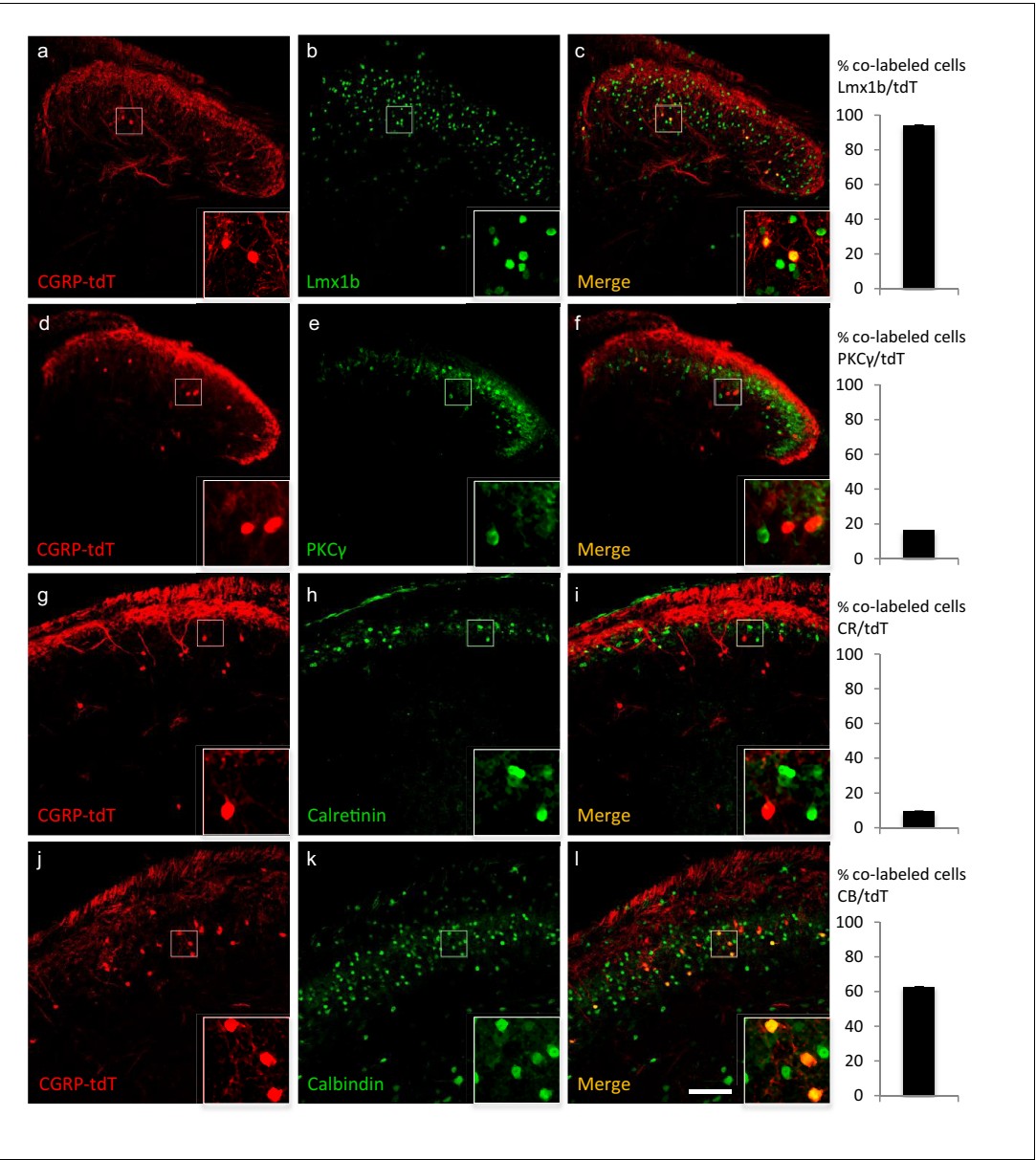

**Figure 2.** CGRP-expressing neurons in the dorsal horn (**a–f**) and nucleus caudalis (**g–l**) are a distinct class of excitatory (Lmx1b+) interneurons. (**a–l**) Immunohistochemistry showed that CGRP-tdTomato fluorescent neurons (red) co-express many markers (green) of excitatory, but not inhibitory (e.g. Pax2, *Figure 2—figure supplement 1*) interneurons in the dorsal horn (**a–f**) and nucleus caudalis (**g–l**). Ninety-eight percent of CGRP-tdTomato neurons co-expressed Lmx1b (**a–c**), 16% co-expressed PKCγ (**d–f**), 9% co-expressed calretinin (**g–i**), and 63% co-expressed calbindin (**j–l**). Insets show higher magnification views of boxed areas in respective images. Graphs illustrate mean percentages ± SEM of CGRP-tdTomato neurons that were double-labeled with the indicated antibody (~100 cells per antibody). Scale bar: 100 μm.

The online version of this article includes the following figure supplement(s) for figure 2:

**Figure supplement 1.** CGRP tdTomato interneurons are Pax2-negative.

Consistent with the concentration of tdTomato-CGRP interneurons in lamina III, particularly notable is that 56% of the *Calca* mRNA-expressing (CGRP) cells double-labeled for *Rora* message (639/ 1134 *Calca* mRNA-expressing cells), a marker of excitatory interneurons in lamina III (*Bourane et al.,*

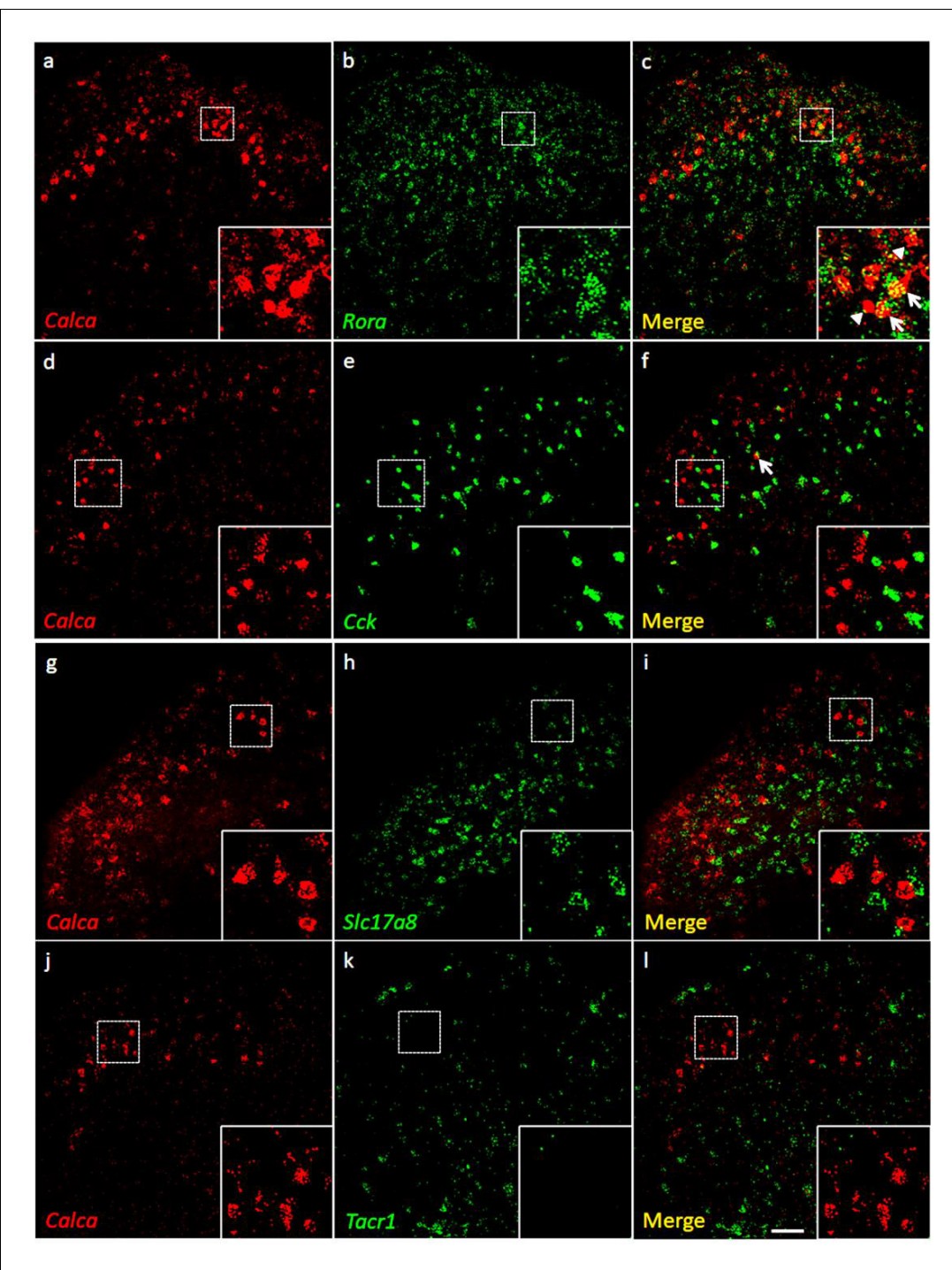

**Figure 3.** Coexpression of *Calca* mRNA with *Rora* mRNA, but with neither *Cck* nor *Tacr1* mRNA. (a-l) Coexpression of *Calca* mRNA (*Calca;* red), with other markers (green) in subsets of dorsal horn (a-c; j–l) and nucleus caudalis (d–i) neurons. Of *Calca*-expressing cells, 56% express *Rora* mRNA (a–c), but only 4.4% express *Cck* mRNA (d–f). Similarly, there was minimal overlap of *Calca* and *Slc17a8*, the gene coding for VGLUT3 (g–i), or *Calca* and *Tacr1*, the gene coding for the NK1 receptor (j–l). Insets show higher magnification images of boxed areas. Scale bar: 100 μm.

*2015b*). Interestingly, however, only 4% co-expressed *Cck* (27/595 *Calca* mRNA-expressing cells), which marks a significant subset of the RORα population (*Liu et al., 2018*). As for other populations of excitatory interneurons, we found minimal overlap with the population that transiently expresses VGLUT3 (examined at P7) (*Peirs et al., 2015*) or others that express Nptx2, BDNF or the NK1 receptor, a marker of many projection neurons. Similar results were found in the dorsal horn of the spinal cord and in the trigeminal nucleus caudalis (viz., dorsal horn of the medulla). We conclude that a substantial portion of the CGRP interneuron population overlaps with a subset of the *Cck*-negative RORα population of lamina III interneurons.

## CGRP interneurons have dorsally directed dendritic arbors and are innervated by VGLUT1-expressing terminals

Despite the very intense tdTomato labeling of the cell bodies of the dorsal horn neurons, it was difficult to distinguish axonal processes from the dense primary sensory neuron-derived CGRP innervation. This was particularly the case when an antibody to tdTomato was used to detect the dorsal horn CGRP neurons. And unfortunately, although the cell body of the intracellularly recorded cells was readily filled with biotin dextran in electrophysiological slice preparations (see below), we never successfully filled dendrites or axons. Therefore, in a separate set of experiments, we first reduced the complement of primary afferent-derived CGRP-derived by making an intrathecal injection of capsaicin, 7 days prior to perfusing the mice (*Cavanaugh et al., 2009*). In addition, tdTomato-immuno-reactivity was revealed with immunoperoxidase staining so that sections could be analyzed by either light or electron microscopy (EM). The results from this approach were both striking and especially informative. *Figure 4* illustrates that the CGRP interneurons have many dorsally-directed, spine-laden dendrites. These dendritic arbors often penetrated lamina II, and some labeled processes appeared to reach lamina I. Nevertheless, despite the capsaicin treatment, the latter were rare and difficult to distinguish from residual primary afferent-derived CGRP.

Based on their remarkably uniform dendritic morphology, the dorsal horn CGRP neurons appear to represent a subpopulation of excitatory, so-called radial interneurons (*Grudt and Perl, 2002*); however, the morphology of the CGRP-expressing radial interneurons differ considerably from those previously described in lamina II. First, the majority of lamina II radial cells have dendrites that arborize ventrally and axons that, if anything, project and collateralize dorsally, occasionally targeting presumptive projection neurons in lamina I. In contrast, not only do the CGRP interneurons have dorsally-directed dendrites, but almost all of their axons project ventrally and/or ventrocaudally. In some instances, we could trace the axons well into the neck of the dorsal horn, including lamina V (*Figure 5* and *Figure 5—figure supplement 1*). Furthermore, EM analysis of these interneurons (*Figure 5*) illustrates that there is significant synaptic input to the soma, dendrites, and spines of the CGRP interneurons. Finally, given the concentration of the CGRP interneurons in lamina III, we assumed that they receive primary afferent input from large myelinated afferents. Indeed when we double-immunostained for tdTomato and VGLUT1, a glutamate transporter that is highly expressed in large myelinated afferents (*Oliveira et al., 2003*), we observed many close appositions of VGLUT1-immunoreactive axon terminals onto the cell bodies and dendrites of the CGRP interneurons (*Figure 4f–j*).

## CGRP-tdTomato interneurons receive low threshold primary afferent input

To confirm that the VGLUT1 appositions indeed mark a monosynaptic input from Aβ afferents to the CGRP-tdTomato interneurons, we prepared transverse lumbar and caudal medullary slices (350–400 µm) from 3-week-old mice for whole-cell patch-clamp recordings. The slices contained large numbers of fluorescent tdTomato-labeled CGRP neurons (*Figure 6a–c*). We first characterized the intrinsic properties of the CGRP-tdTomato neurons by inducing depolarizing current steps. The CGRP-tdTomato neurons in the dorsal horn and nucleus caudalis showed mostly delayed firing patterns, consistent with their excitatory and radial phenotype (delayed 19, tonic 1, reluctant 2, single 2, no response 3, *Figure 6—source data 1* table). In some preparations we stimulated an attached dorsal root. At near threshold stimulation intensities (10 Hz), we recorded a very short latency component, which likely corresponds to a monosynaptic Aβ-fiber input. Of five cells recorded in three mice, all received monosynaptic Aβ input. In two additional mice, we recorded from four cells that responded

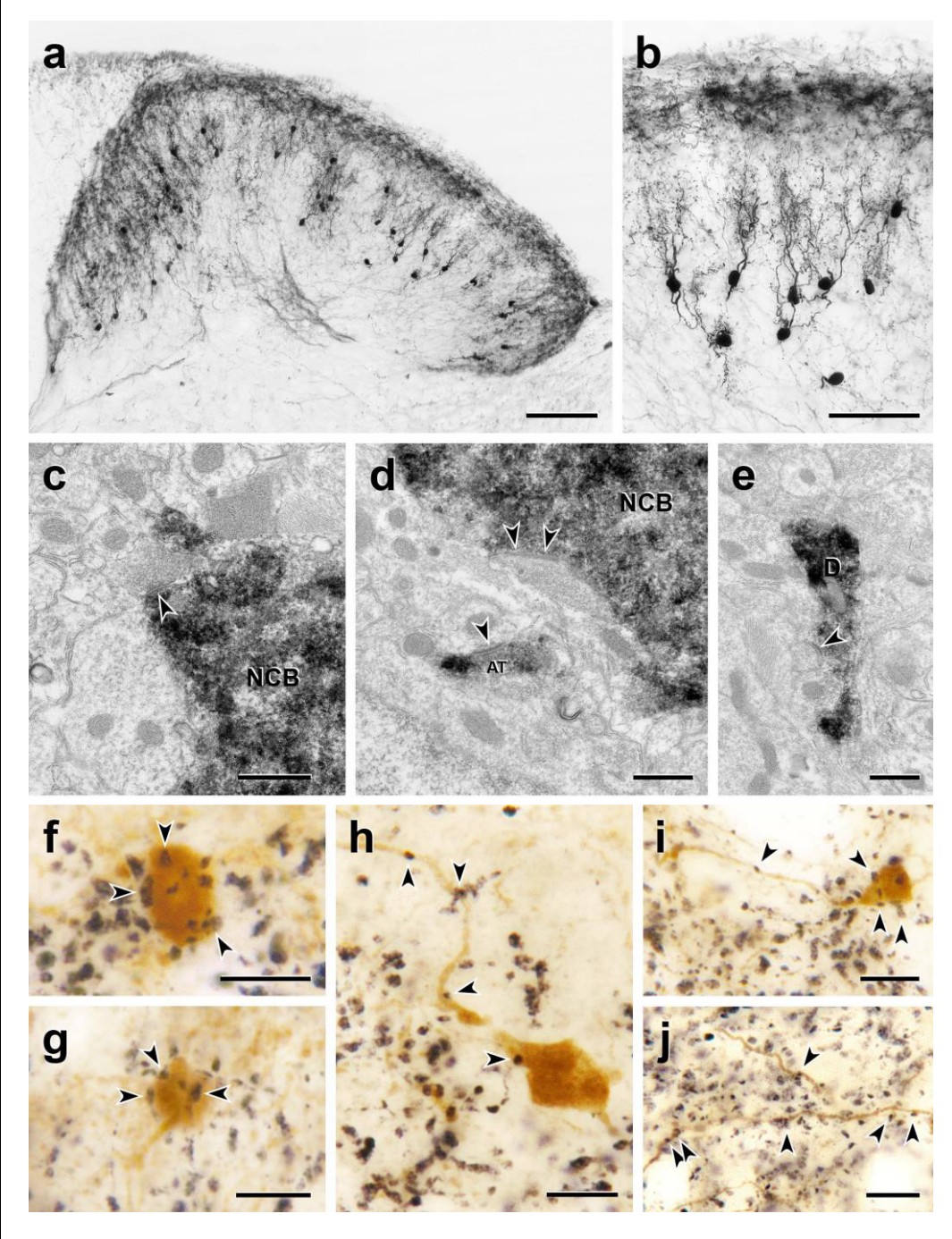

**Figure 4.** Morphology and VGLUT1 innervation of dorsal horn CGRP interneurons. (**a, b**) Most tdTomato-immunoreactive CGRP interneurons (black) are located in lamina III and have a relatively uniform morphology with many spiny, dorsally projecting dendrites. Scale bars: 100 μm in a; 50 μm in **b**. (**c-e**) Electron microscopic analysis revealed unlabeled host synapses (arrowheads) presynaptic to the cell bodies (NCB in **c** and **d**) and dendrites (D in **e**) of tdTomato-immunoreactive (black) CGRP interneurons. **d** also shows an asymmetric presynaptic input (AT) from a presumptive CGRP interneuron to an unlabeled host dendrite. (**f – j**) Black VGLUT1-immunoreactive varicosities form close appositions (arrowheads) with the cell bodies (**f and g**) and dendrites (**h – j**) of brown tdTomato-immunoreactive CGRP interneurons. Scale bars: 500 nm in **c – e**, 10 μm in **f – j**.

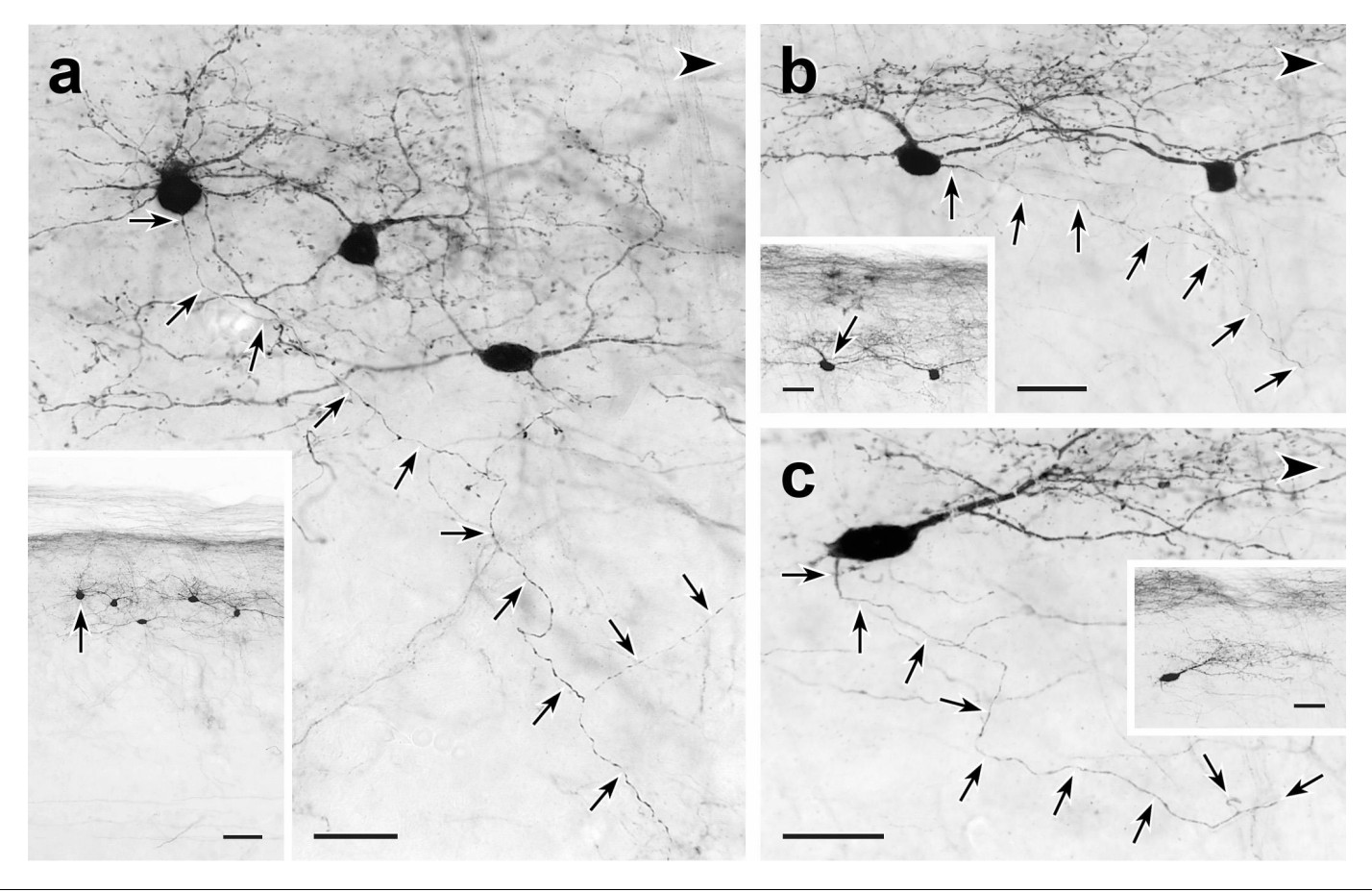

**Figure 5.** Trajectories of axons of CGRP-tdTomato interneurons. tdT-immunoreactive CGRP interneurons (black) in 50 μm parasagittal sections from the lumbar dorsal horn of CGRP-tdTomato mice in which an intrathecal injection of capsaicin reduced primary afferent-derived CGRP. The CGRP-tdTomato neurons have spiny, dorsally directed dendrites and their axons (arrows) course ventrally and often caudally (large arrowhead) (*Figure 5—figure supplements 1* and *2*). Arrows in insets indicate location of the neurons whose axons are shown in a, b and c. Scale bars: 20 μm in a–c, 50 μm in inset a, 20 μm in insets b and c.

The online version of this article includes the following figure supplement(s) for figure 5:

**Figure supplement 1.** Dorsal horn CGRP interneurons have ventrally directed axons.

**Figure supplement 2.** Radial morphology of the CGRP-tdTomato interneurons revealed after AAV injection.

to dorsal root stimulation, but we could not unequivocally establish whether they received a monosynaptic input (*Figure 6d–e*). Overall, the intrinsic properties of neurons recorded from lumbar dorsal horn (22 cells, eight mice) and nucleus caudalis (5 cells, two mice) were comparable (see *Figure 6—source data 1* table). Taken together, we conclude that the predominant (monosynaptic) input to the CGRP interneurons derives from low threshold (Aβ) mechanoreceptors.

## Blocking tonic inhibition increases excitability of the CGRP interneurons

In a separate set of experiments, we specifically sought evidence that the neurons, under baseline conditions, are under inhibitory control. To this end, cells were patched and then rheobase determined, before and after application of a combination of bicuculline and strychnine. *Figure 7a–d* illustrate that concurrent blocking of the GABA and glycine receptors significantly reduced rheobase, from $46.0 \pm 7.4$ pA before antagonist treatment to $31.0 \pm 4.9$ pA after antagonist treatment (two-tailed, paired T-test; p=0.0005, n = 25). Of the 25 neurons studied, rheobase decreased in 21, increased in one and did not change in 3. *Figure 7c* shows that application of bicuculline and strychnine to neurons in which current was maintained 10 pA below rheobase also generated action potentials. *Figure 7e* shows that resting membrane potential also showed a significant

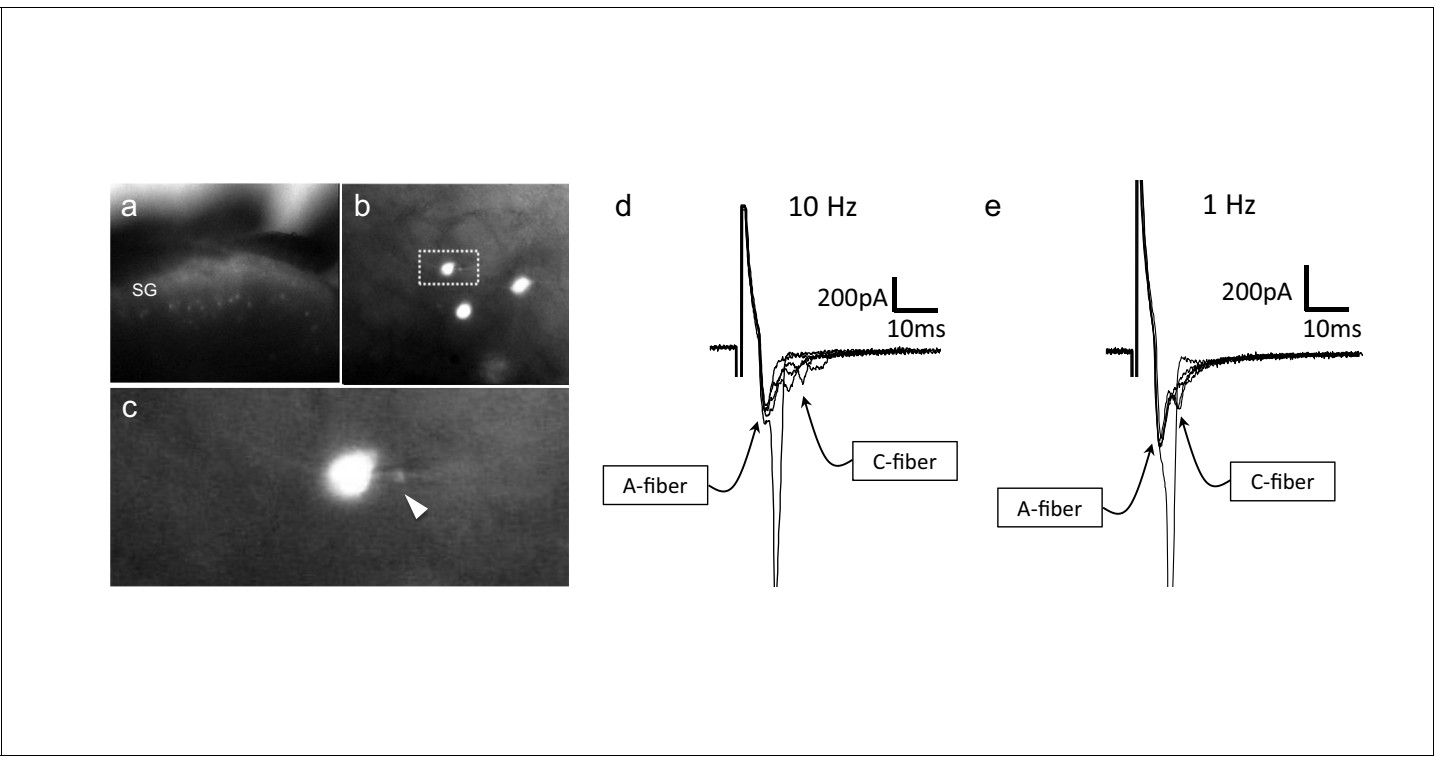

**Figure 6.** CGRP-tdTomato interneurons receive low threshold sensory inputs. Low (**a**) and high (**b**) magnification micrographs of endogenous fluorescent CGRP-tdTomato neurons in a spinal cord slice. The boxed neuron in (**b**) is shown at high magnification in **c**; arrowhead points to the recording pipette in a whole cell configuration. (**d,e**) Responses of CGRP-tdTomato interneuron to dorsal root stimulation at 10 Hz (**d**) or 1 Hz (**e**). An early, persistent component likely corresponds to a monosynaptic A-fiber input. The late component, with variable latency and failures, likely reflects polysynaptic C-fiber input (electrophysiological properties: *Figure 6—source data 1* table).

The online version of this article includes the following source data for figure 6:

**Source data 1.** Electrophysiological properties of CGRP-tdTomato interneurons in the dorsal horn and nucleus caudalis.

depolarization after application of the GABA and glycine receptor antagonists, from −53.8 ± 1.8 mV before antagonist treatment to −49.1 ± 1.7 mV after antagonist. Taken together, these results demonstrate that the CGRP interneurons, under resting conditions, are under tonic inhibitory control.

## Mechanical stimuli only activate CGRP neurons (induce Fos expression) in a nerve injury setting

To provide a global activity measure of the stimuli that engage the CGRP interneurons, we first monitored Fos expression using a battery of noxious and innocuous stimuli. As expected, a unilateral injection of dilute formalin into the cheek (10 µl of 2% formalin, *Figure 8—figure supplement 1c*) or a unilateral hindpaw injection of capsaicin (*Figure 8—figure supplement 2a–b*), produced considerable Fos immunolabeling of dorsal horn neurons, but not of the CGRP-tdTomato interneurons (*Figure 8—figure supplements 1c* and *2a–b*). Unexpectedly, however, selectively engaging non-nociceptive afferents by having the animal walk for 90 min on a rotarod, which provokes considerable Fos in laminae III and IV (*Neumann et al., 2008*), did not induce Fos expression in the CGRP interneurons (*Figure 8—figure supplement 1a*). The same was true for brushing of the cheek, another innocuous stimulus that activates Aβ afferents (*Figure 8*). Finally, although CGRP is strongly implicated in the generation of migraine, largely but not exclusively via its peripheral vasodilatory action (*Brain et al., 1985*), systemic injection of nitroglycerin, which triggers migraine in humans and profound mechanical hypersensitivity in animals (*Bates et al., 2010*), did not induce Fos in the CGRP interneurons (*Figure 8—figure supplement 1b*).

We conclude that despite our electrophysiological evidence that Aβ afferents engage the CGRP interneurons, there does not appear to be sufficient input to activate these cells under natural innocuous mechanical stimulus conditions in uninjured mice (5.3%, 5/88 tdTomato cells; *Figure 8a*). We,

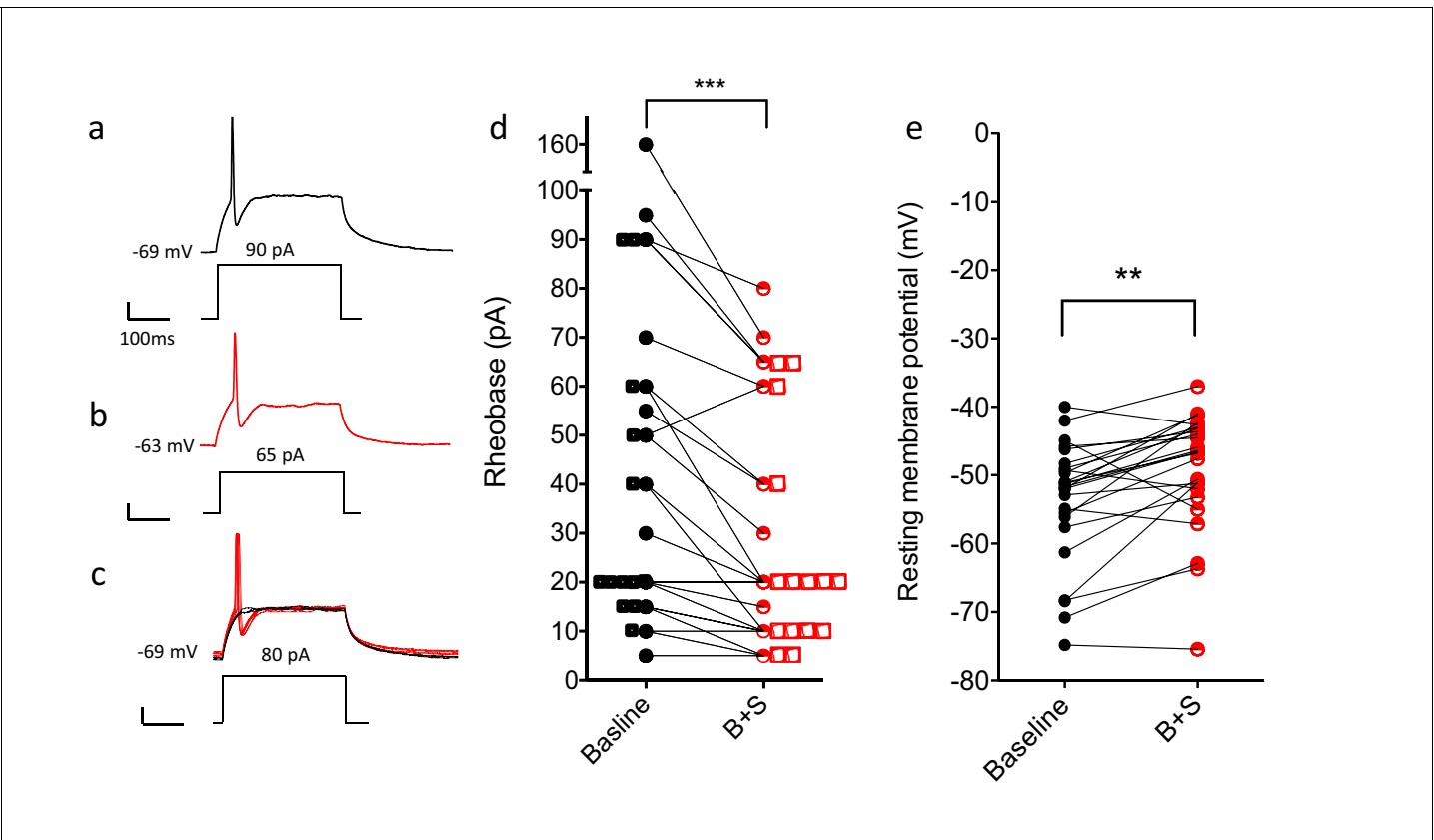

**Figure 7.** Blocking GABA and glycine receptors increases the excitability of the CGRP interneurons. Representative traces show current evoked action potential before (**a**) and after (**b**) bicuculline and strychnine. (**c**) Administration of antagonists when current application was 10 pA below rheobase threshold (black) also induced an action potential (red). (**d**) Rheobase values before and after antagonist treatment (n = 25; two-tailed, paired T-test, p=0.0005). (**e**) Resting membrane potentials before and after bicuculline (B) and strychinine (S) treatment (n = 25).

therefore, next asked whether an injury state would render the CGRP interneurons more responsive to an innocuous stimulus. In fact six days after inducing the spared nerve injury (SNI) model of neuro-pathic pain, we found that brushing the ipsilateral paw evoked Fos expression in 50% (53/110 tdTo-mato cells) of the dorsal horn CGRP interneurons (*Figure 8c and d*). Importantly, although we recorded significant dorsal horn Fos expression in nerve-injured mice without brushing (*Figure 8b*), no Fos expression occurred in the CGRP interneurons (3%; 6/205 tdTomato cells). We conclude that activation of the CGRP interneurons only occurs when the innocuous input, which could include con-tact of the plantar surface of the paw with the ground (*Liu et al., 2018*), engages the interneurons in the setting of nerve injury.

## Dorsal horn CGRP interneurons contribute to mechanical hypersensitivity in vivo

As electrical stimulation of the dorsal root at Aβ intensity readily excites the CGRP interneurons, the inability of brush stimulation to activate the neurons in the absence of injury was surprising. The dis-crepancy may reflect the fact that dorsal root stimulation involves a synchronous activation of many primary sensory neurons. In contrast, natural stimuli (e.g. brushing or walking on a rotarod) trigger an asynchronous afferent drive. However, as brushing was effective in the nerve injury setting, we hypothesized that a central sensitization rendered the CGRP neurons hyperexcitable. To test this hypothesis, we asked whether a different mode of activation, namely chemogenetic (direct) activa-tion of the CGRP interneurons, could generate behaviors indicative of mechanical allodynia, compa-rable to what is observed in response to innocuous mechanical stimuli in the setting of nerve injury.

In these studies, we used an intersectional approach to target expression of a Designer Receptor Exclusively Activated by Designer Drugs (DREADD) selectively in the CGRP interneurons. To this

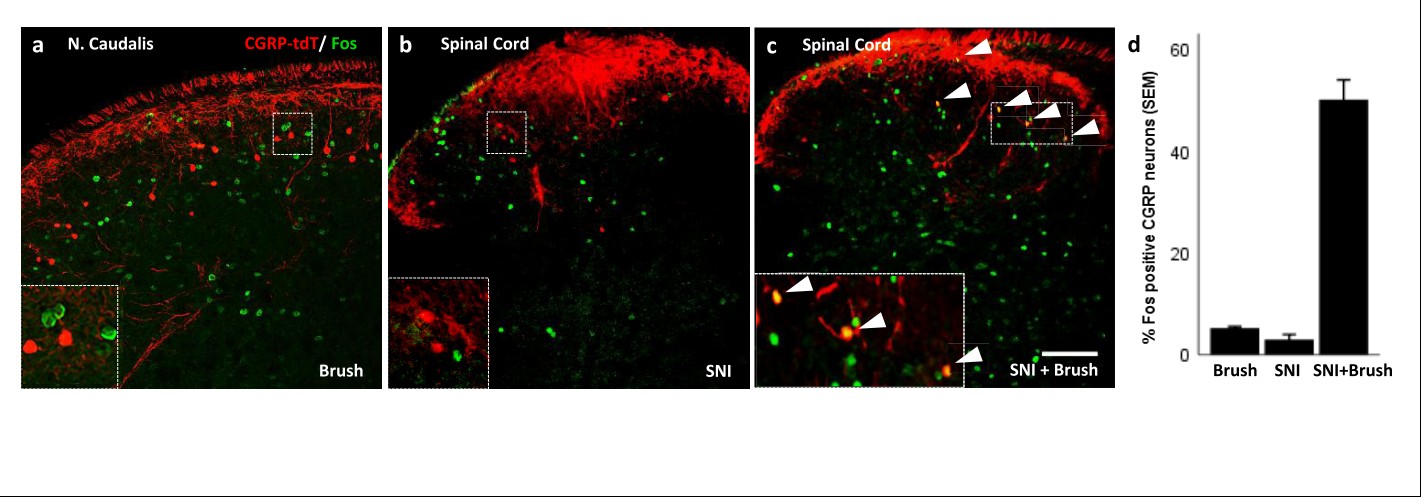

**Figure 8.** Peripheral innocuous stimuli activate CGRP interneurons but only after spared nerve injury (SNI). (**a**) Fos-immunoreactive neurons in nucleus caudalis after brushing the cheek of a naive uninjured mouse. (**b**) Fos expression in the lumbar dorsal horn 6 days after SNI without additional peripheral stimulation. (**c**) Fos expression in the lumbar dorsal horn 6 days after SNI with additional brush stimulation of the hindpaw. Insets: high-magnification images of the boxed areas in the respective micrographs. Arrowheads indicate double-labeled cells. Scale bar: 100 µm. (**d**) Mean percentages ± SEM of CGRP-tdTomato neurons that are Fos-immunoreactive in the different conditions (*Figure 8—figure supplements 1* and *2*). The online version of this article includes the following figure supplement(s) for figure 8:

**Figure supplement 1.** Neither noxious nor innocuous stimuli induce Fos expression in CGRP-tdTomato interneurons in control mice.

**Figure supplement 2.** Capsaicin does not activate CGRP-tdTomato interneurons in the lumbar spinal cord or trigeminal nucleus caudalis.

end, we crossed the *Calca^creER* mice to a FLPo mouse line, driven by the *Lbx1* gene. The latter gene is only expressed in neurons of dorsal spinal cord and hindbrain, but not in sensory neurons of the DRG (*Bourane et al., 2015b*). We then made a unilateral microinjection of an adenoassociated virus (AAV) expressing a Cre and FLPo-dependent DREADD (hM3Dq) into the dorsal horn of the *Calca^creER*/FLPo mice. Four weeks later, we evaluated the behavioral effects of a systemic injection of CNO, which activates the DREADD.

We first established that there was no constitutive effect of virus infection. Thus, CNO injection, compared to saline, did not alter the latency to fall from an accelerating rotarod (*Figure 9c*). Furthermore, baseline von Frey mechanical thresholds of the DREADD-expressing mice, measured prior to injection of CNO, did not differ from mice injected with the AAV-GFP virus. In distinct contrast, *Figure 9* shows that CNO injection in the experimental group produced a significant reduction of von Frey threshold of the ipsilateral hindpaw, compared to baseline or to saline-injected mice (*Figure 9b*). Mechanical thresholds did not change from baseline in the AAV-GFP control animals, whether they received saline or CNO (Repeated Measures Two-way ANOVA, $F_{(1,20)}$=6.964, p=0.012, interaction effect between DREADD group and CNO treatment). The groups contained the same numbers of males and females (DREADD animals: 8 of each; GFP controls: 3 of each), but there was no significant interaction between sex and treatment (CNO versus saline). Nor did factoring in sex reduce the error ($R^2$) in the full Repeated Measures Two-way ANOVA. Consistent with a contribution of CGRP-expressing interneurons to mechanical sensitivity, mice in which we ablated selectively the CGRP-expressing spinal cord interneurons with a virally derived caspase (*Figure 10a* and *Figure 10—figure supplement 1*) exhibited significantly higher mechanical thresholds than did control mice (*Figure 10b*). On the other hand, and somewhat unexpectedly, the mechanical hypersensitivity produced in the SNI model of neuropathic pain was not altered by the ablation.

Lastly, we evaluated heat and cold responsiveness after CNO injection. Neither latency to withdraw the hindpaw to noxious heat in the Hargreaves test (n = 16; *Figure 9d*) nor time spent paw lifting after exposure of the plantar surface of the hindpaw to a cold (acetone) stimulus (n = 11; *Figure 9e*), differed when comparing CNO and control saline injection (p>0.05, Students T-test and Wilcoxon Signed Ranks Test, respectively). Similarly, responses to noxious heat (Hargreaves; *Figure 10c*) or cold (acetone; *Figure 10d*) stimuli were unchanged after caspase-mediated ablation

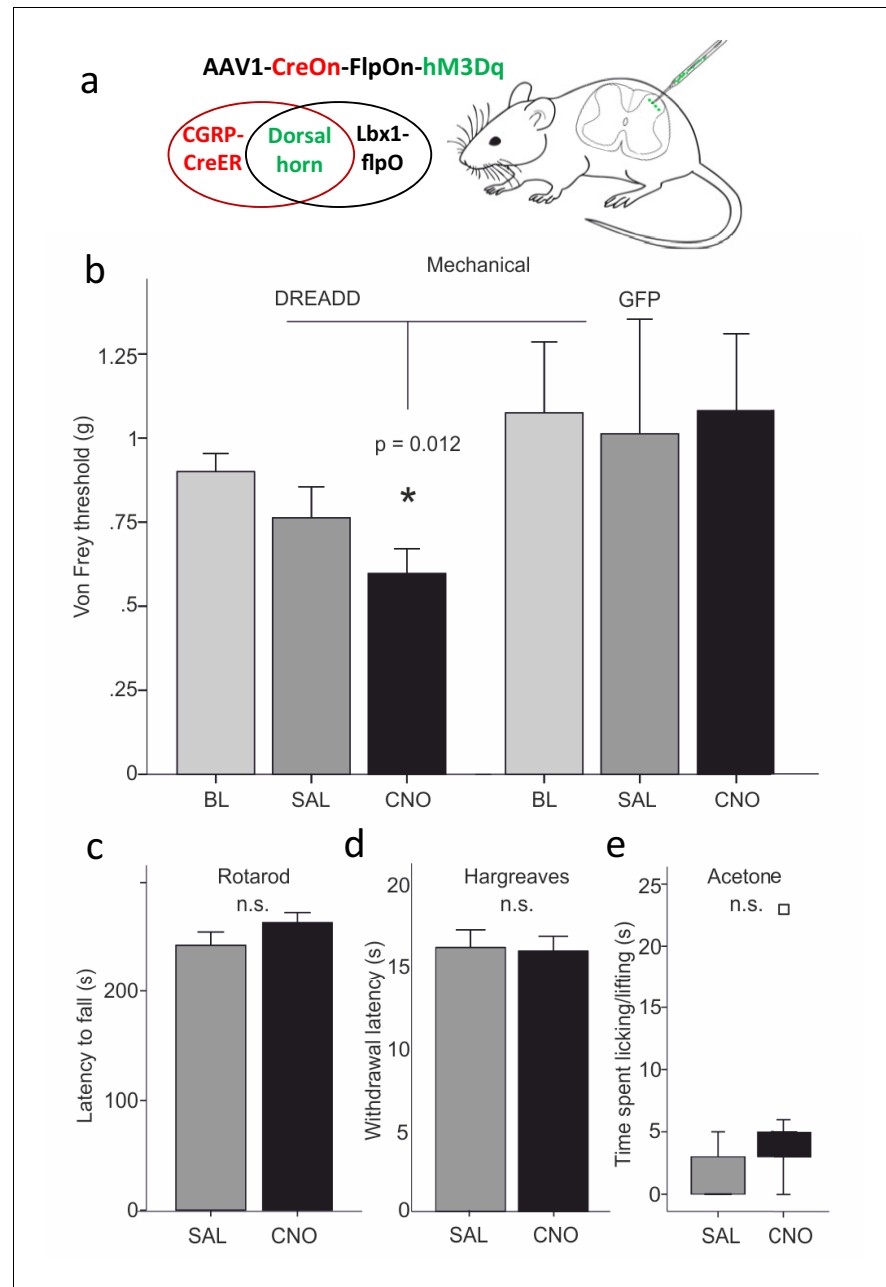

**Figure 9.** Dorsal horn CGRP interneurons contribute to mechanical sensitivity in vivo. (**a**) $Calca^{CreER}$ mice were crossed to an Lbx1-driven FLPo mouse line, which restricts Cre expression to Lbx1-expressing neurons in the dorsal spinal cord and hindbrain. We then injected a Cre and FLP-dependent DREADD (hM3Dq) virus (AAV1-CreOn-FlpOn-hM3Dq) or a GFP-expressing AAV into the lumbar dorsal horn. (**b**) Baseline (BL) von Frey mechanical thresholds of the DREADD-expressing mice (n = 16; light grey bars) did not differ from baseline thresholds of mice injected with the AAV-GFP (GFP) control virus (n = 6). In contrast, CNO injection significantly reduced von Frey thresholds (CNO, black bars) of the ipsilateral hindpaw in the DREADD-injected mice, compared either to their baseline, to the GFP controls or to saline (SAL; light grey bars)-injected mice (Repeated measures Two-way ANOVA, p=0.012). Neither latency to fall from a rotarod (**c**), withdrawal to noxious heat in the Hargreaves test (**d**), nor time spent paw lifting after exposure of the paw to a cold stimulus (acetone) (**e**) differed when comparing CNO and the control saline injection (p>0.05, Students T-test and Wilcoxon Signed Ranks Test, respectively). Square in (**e**) indicates an outlier.

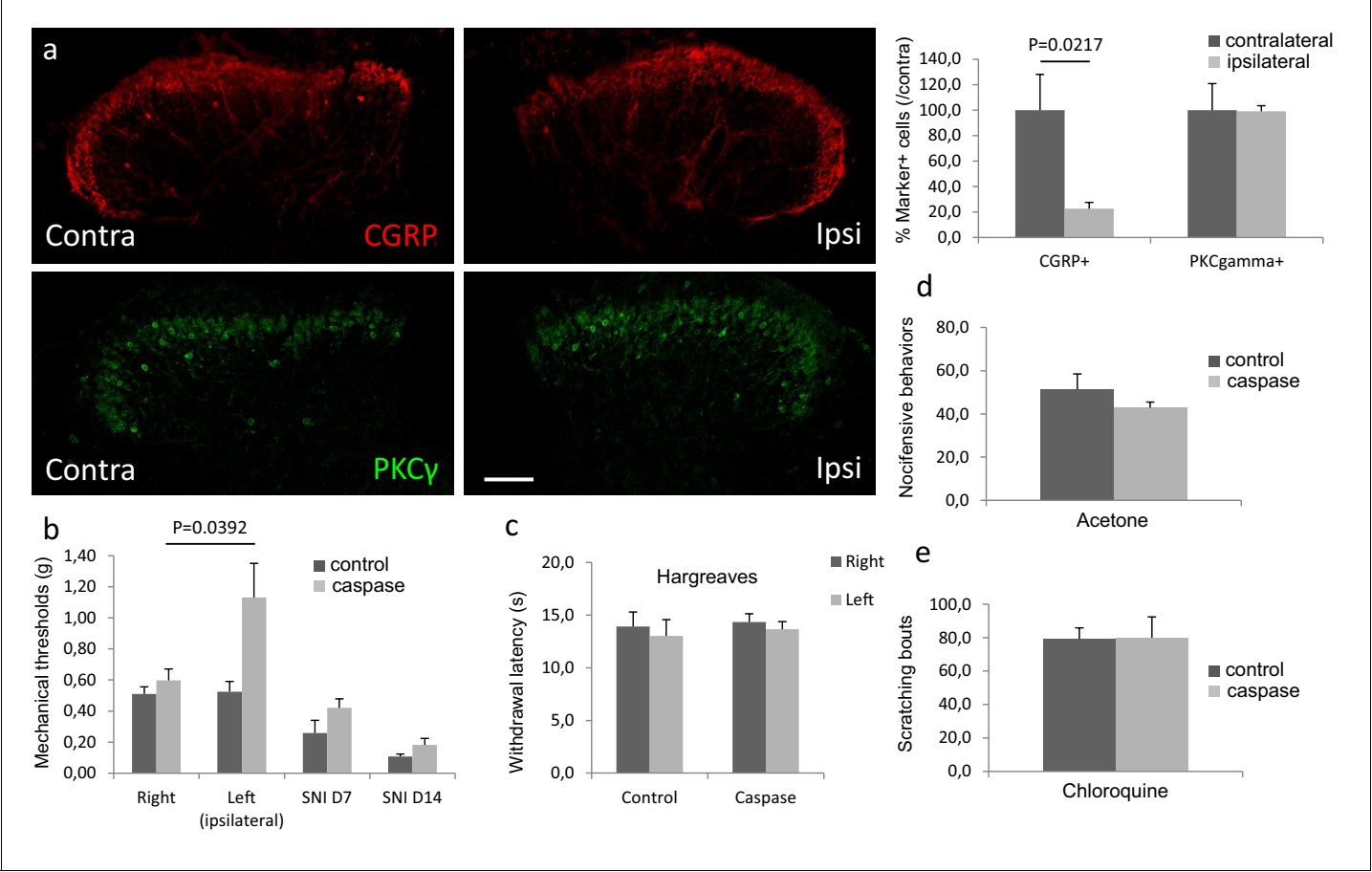

**Figure 10.** Ablation of dorsal horn CGRP interneurons decreases mechanical sensitivity in vivo. (a) The number of CGRP (red)-expressing interneurons was significantly decreased 4 weeks after injection of a Cre-dependent caspase-expressing viral vector into the superficial dorsal horn of *Calca^CreER*/tdTomato mice. In contrast, the number of PKCγ (green)-expressing interneurons did not change. Scale bar: 100 μm. (b) The von Frey mechanical thresholds were significantly higher ipsilateral to the injection side (left) after the CGRP interneuron ablation (n = 8; light grey bars), compared to the contralateral (right) side. In contrast, the von Frey mechanical thresholds did not differ in control mice (n = 6; dark gray bars). Nevertheless, CGRP-ablated and control mice exhibited similar levels of mechanical hypersensitivity 7 and 14 days after SNI. (c–e) Ablation of spinal cord CGRP-expressing interneurons did not change the withdrawal latencies in the Hargreaves test (c), the number of paw lifts in response to a cold stimulus (acetone) (d) or the number of scratching bouts evoked by a subcutaneous calf injection of chloroquine (unpaired Students T-test).

The online version of this article includes the following figure supplement(s) for figure 10:

**Figure supplement 1.** The number of CGRP (red)- or PKCγ (green)-expressing interneurons did not change 4 weeks after injection of saline into the superficial dorsal horn of *Calca^CreER*/tdTomato mice (unpaired Students T-test).

of the CGRP-expressing interneurons. We conclude that direct and likely synchronous activation of the CGRP interneurons produces a selective mechanical hypersensitivity, mimicking the mechanical allodynia observed in response to low threshold (Aβ) mechanical stimulation (brush) in the setting of nerve injury.

## CGRP interneurons and itch

Based on their single cell transcriptome analysis, Häring and colleagues (*Häring et al., 2018*) concluded that several populations of dorsal horn excitatory neurons that express *Calca* mRNA co-express gastrin-releasing peptide (GRP), a peptide linked to dorsal horn circuits that drive itch-provoked scratching (*Albisetti et al., 2019*; *Sun and Chen, 2007*). To confirm this, we performed double in situ hybridization for *Calca* and *Grp*. Although the *Grp* interneurons predominated in a band just dorsal to the *Calca* interneurons, consistent with our previous report (*Solorzano et al., 2015*), we did find several instances of co-localization of *Calca* mRNA and *Grp* mRNA. Interestingly, however, when using immunohistochemistry, we found almost no overlap of GRP and CGRP in a double

transgenic GRP-GFP/CGRP-tdTomato mouse line (*Figure 11a–d*). This difference is likely related to the fact that neurons labeled in the reporter mouse constitute less than half of the *Grp* mRNA-positive population (*Solorzano et al., 2015*; *Dickie et al., 2019*). Despite these discordant findings, we also examined the pattern of Fos expression provoked by injection of chloroquine (CQ), a strong pruritogen, into the cheek or hindpaw. To prevent scratching-induced Fos, the CQ injections were performed in anesthetized mice. As *Figure 11e–f* illustrates, despite considerable chloroquine-induced Fos expression, we found only an occasional double-labeled neuron. Furthermore, the number of scratching bouts induced by a subcutaneous calf injection of 100 μg chloroquine was comparable between control and CGRP-ablated mice (*Figure 10e*). We conclude that the CGRP interneurons, despite some overlap with GRP, likely do not transmit chemical itch, a finding consistent with the effects of deleting RORα (*Bourane et al., 2015b*). Whether the CGRP interneurons are engaged in conditions in which mechanical stimulation can trigger itch (alloknesis) remains to be determined.

## Discussion

Despite overwhelming evidence that primary sensory neurons are the predominant source of dorsal horn CGRP, here we describe a morphologically uniform population of dorsal horn CGRP-expressing interneurons. Many of these interneurons correspond to the *Cck*-negative subset of the RORα population in lamina III of the dorsal horn and trigeminal nucleus caudalis, are excitatory and are activated by electrical stimulation of non-nociceptive, Aβ primary afferents. In contrast to the *Cck*-expressing subset of RORα neurons, and despite their location in the so-called, low threshold mechanoreceptive recipient zone of the dorsal horn (*Abraira et al., 2017*), the CGRP interneurons do not express Fos in response to natural Aβ-mediated, innocuous mechanical stimulation (brushing or walking on a rotarod). We hypothesize that this reflects competition with the ongoing inhibition of these neurons (see below). As for the RORα population, the CGRP interneurons do not respond to noxious chemical stimulation. Even peripheral nerve injury, without superimposed stimulation, did not activate these neurons. On the other hand, brush stimulation in the nerve injury setting did activate the CGRP interneurons. Furthermore, and consistent with a limited contribution of these neurons in the setting of nerve injury, ablation of CGRP interneurons did not influence the magnitude of mechanical allodynia that develop following peripheral nerve injury. Interestingly, however, brush stimulation in the nerve injury setting did induce Fos in the CGRP interneurons. This distinction suggests that unless these neurons are rendered hyperexcitable, as occurs after nerve injury, only synchronous afferent input or direct neuronal sensitization (e.g. by DREADD activation) is sufficient to engage the circuits in which the CGRP interneurons participate. Consistent with this conclusion, chemically provoked (chemogenetic) synchronous activation of these neurons produced a significant mechanical hypersensitivity and conversely their ablation increased mechanical thresholds. Based on the predominant ventrally directed axonal arbors of these interneurons we suggest that the dorsal horn CGRP interneurons contribute either to ascending circuits originating in deep dorsal horn or to the reflex circuits in baseline conditions, but not in the setting of nerve injury. The fact that nerve injury-induced mechanical hypersensitivity persisted after ablation of the CGRP interneurons undoubtedly reflects the major contribution of other mechanosensitive afferents and dorsal horn interneurons. Indeed, we previously reported that the MrgprD subpopulation of sensory neurons is an important driver of the nerve-injury induced mechanical sensitivity (*Cavanaugh et al., 2009*) and these afferents target interneurons located dorsal to the predominant band of CGRP interneurons.

RNA-Seq analyses have now defined at least 15 subsets of excitatory interneurons and 15 subsets of inhibitory neurons in the dorsal horn of the spinal cord (*Häring et al., 2018*; *Sathyamurthy et al., 2018*). Ablation, optogenetic and chemogenetic studies further characterized those classes based on functional properties. Of note, an increasing number of dorsal horn interneurons that 'gate' mechanical pain have been identified. These include neurochemically distinct excitatory interneuron populations: transient VGLUT3, somatostatin, RORα, calretinin, and Tac1 (*Bourane et al., 2015a*; *Cheng et al., 2017*; *Duan et al., 2014*; *Huang et al., 2019*; *Peirs et al., 2015*; *Petitjean et al., 2019*) and distinct inhibitory interneuron populations: dynorphin, calretinin, parvalbumin, and enkephalin (*Boyle et al., 2019*; *Duan et al., 2014*; *François et al., 2017*; *Petitjean et al., 2019*; *Petitjean et al., 2015*). The CGRP-expressing interneurons define yet another population of dorsal horn interneurons that contributes to spinal cord processing of mechanical inputs. Interestingly,

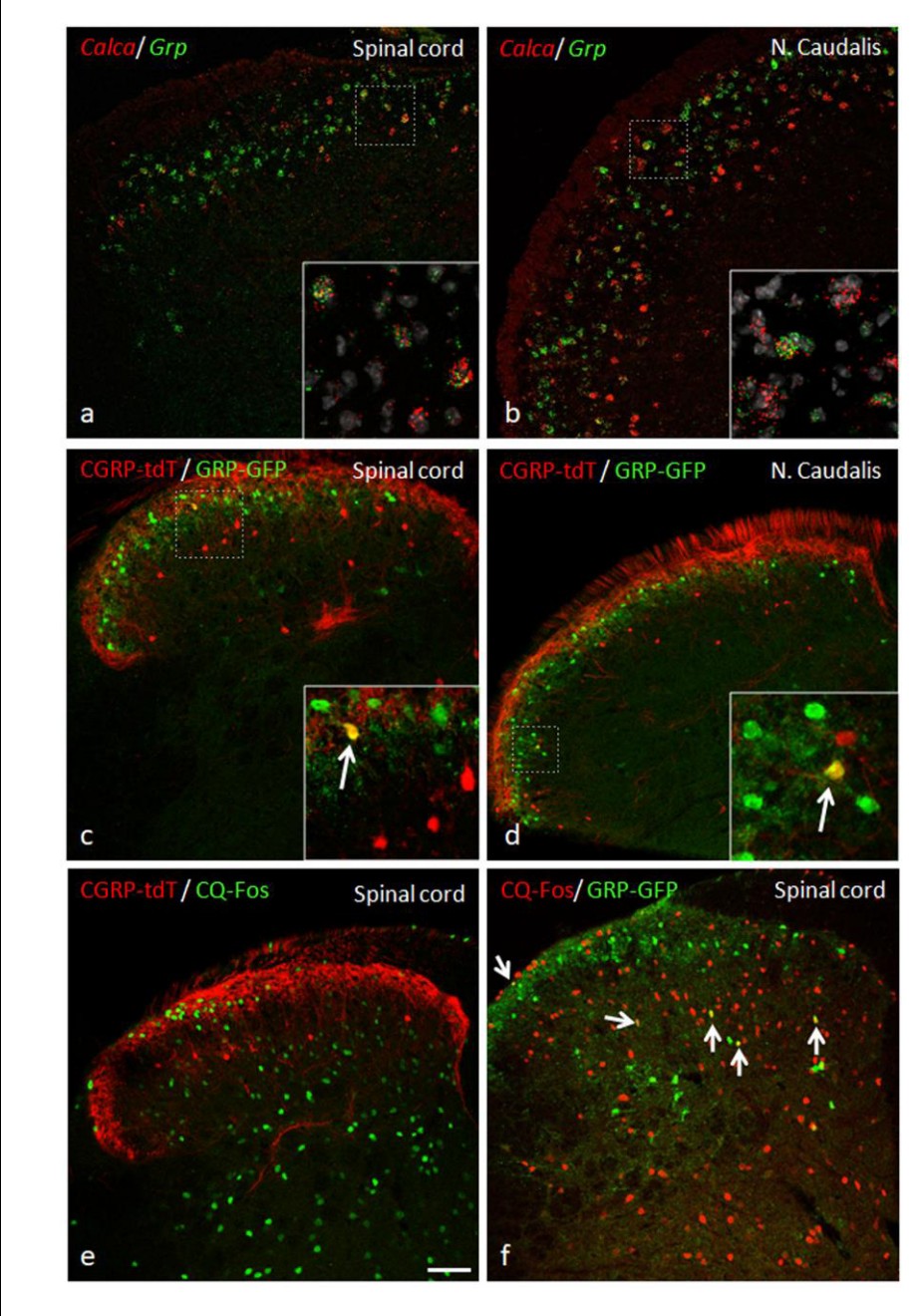

**Figure 11.** GRP, CGRP, and pruritogen-evoked Fos expression. Double in situ hybridization for tdTomato (red) and *Grp* (green) illustrates considerable mRNA co-expression in neurons of the dorsal horn (**a**) and nucleus caudalis (**b**) of CGRP-tdTomato mice. In contrast, immunocytochemical localization of GRP and tdTomato in a tamoxifen-treated *Calca*^CreER/tdTomato mouse that was crossed with a GRP-GFP reporter mouse revealed only occasional double labeling (arrow in inset) in the dorsal horn (**c**) or nucleus caudalis (**d**). Consistent with this minimal overlap, Fos expression in tdTomato-labeled CGRP interneurons was rare in response to a hindpaw injection of chloroquine (CQ; **e**). In contrast, many GRP-GFP interneurons were immunostained for Fos in response to CQ (arrows in **f**). As the mice were anesthetized the CQ-induced Fos was scratching-independent. Scale bar: 100 μm.

there is a striking laminar organization of these molecularly distinct populations of interneurons. For example, the transiently expressing VGLUT3 population is located ventral to the CGRP interneurons, receives low-threshold mechanoreceptive input and their chemogenetic activation also enhances mechanical sensitivity (*Cheng et al., 2017*; *Peirs et al., 2015*). Dorsal to the CGRP interneuron are PKCγ and calretinin excitatory interneurons that contribute to nerve injury induced mechanical allodynia (*Malmberg et al., 1997*; *Neumann et al., 2008*; *Peirs et al., 2015*; *Petitjean et al., 2019*; *Smith et al., 2019*).

To what extent these mechanically driven neuronal populations are interconnected or whether they represent parallel, independent circuits activated under different mechanical pain conditions (e.g. naive vs injury vs inflammation) remains to be determined. Here, the unique morphology of the CGRP interneurons is instructive. In contrast to many of the interneuron populations whose axons arborize longitudinally (e.g. PKCγ cells) or dorsally (e.g. calretinin cells), the CGRP interneurons have ventrally-directed axons. In some respects, the CGRP interneurons resemble the lamina II radial cells described by *Grudt and Perl, 2002* in the mouse, many of which are nociceptive, and the lamina III interneurons demonstrated in Golgi preparations in the cat and primate (*Beal and Cooper, 1978*; *Maxwell, 1985*). The fact that the CGRP interneurons show delayed firing patterns is also consistent with the properties of excitatory lamina II radial cells (*Dickie et al., 2019*; *Grudt and Perl, 2002*; *Punnakkal et al., 2014*; *Yasaka et al., 2010*). Surprisingly, despite their dorsal dendrites, which extend into lamina II, where many nociceptive afferents terminate, we found no evidence that the CGRP interneurons are activated by acute noxious inputs (capsaicin or formalin). On the other hand, we did detect an occasional polysynaptic input following synchronous electrical stimulation of primary afferent C fibers. Most importantly, compared to the lamina II radial cells, we recorded much more extensive ventral axon trajectories of the CGRP interneurons, which suggests that these interneurons engage very different circuits in the dorsal and potentially ventral horn. In this regard, the CGRP interneurons are distinct from the calretinin interneurons that target lamina I projection neurons (*Petitjean et al., 2019*).

An RNA sequencing study of dorsal horn interneurons demonstrated expression of *Calca*, the gene that encodes CGRP, in different clusters of neurons (*Häring et al., 2018*), including several that express *Rora*, the gene that encodes RORα. Consistent with those results, our in situ hybridization studies found extensive co-expression of *Calca* and *Rora*. In fact, almost 55% of the CGRP interneurons co-express RORα message and there are significant similarities in their anatomical and functional properties (*Bourane et al., 2015b*). Specifically, the majority of RORα interneurons are excitatory and approximately 1/3 has a radial morphology, with ventrally arborizing axons. Furthermore, both the CGRP and RORα interneurons receive a monosynaptic Aβ afferent input and interestingly, despite the lack of response to capsaicin, some neurons in both populations receive a polysynaptic A delta and C input. Consistent with the report that deletion of the RORα population did not influence itch (*Bourane et al., 2015b*), and despite some overlap of the CGRP and GRP subsets of interneurons, we found that pruritogens did not activate (induce Fos) in the CGRP interneurons and ablation of CGRP interneurons did not influence scratching in response to exogenous pruritogens. There are, however, some striking differences between the RORα and CGRP interneurons. For example, although a majority of the RORα interneurons co-express *Cck*, the CGRP interneurons rarely do. Furthermore, whereas RORα interneurons are activated by innocuous mechanical stimuli (e.g. brushing) in both naive and injured conditions, the CGRP interneurons respond to innocuous stimuli only in the setting of nerve injury.

To our knowledge, the CGRP interneurons represent the first class of excitatory interneurons in lamina III that are unresponsive to innocuous mechanical stimulation under basal conditions despite receiving a monosynaptic Aβ input. One possibility is that the CGRP interneurons are tonically inhibited under normal conditions, which is consistent with our electrophysiological recordings showing bicuculline-mediated facilitation of these interneurons. Reduction of these inhibitory inputs in the setting of injury (*Torsney and MacDermott, 2006*) would render the neurons responsive to an innocuous stimulus (e.g. brush). In turn, the ventrally directed axons of these interneurons could drive reflex withdrawal circuits, which is consistent with increased mechanical thresholds in the CGRP-interneuron ablated mice, and/or engage ascending nociceptive pathways located in deep dorsal horn. The fact that DREADD-mediated direct activation of many CGRP interneurons lowered mechanical withdrawal thresholds is consistent with that hypothesis. In other words, we suggest that sensitization of these neurons is critical to mechanisms that underlie Aβ-mediated mechanical

allodynia in the setting of nerve injury. Interestingly, *Lu et al., 2013* provided evidence for convergence of a primary afferent-derived Aβ and a tonic glycinergic inhibitory input to PKCγ interneurons, some of which we found express CGRP. Loss of this glycinergic inhibition allowed Aβ input to access lamina I nociceptive circuits. Other studies demonstrated a comparable outcome, in this case by a presynaptic glycinergic inhibition of non-nociceptive inputs to superficial dorsal horn neurons (*Sherman and Loomis, 1996*). Furthermore, *Imlach et al., 2016* proposed that decreased glycinergic inhibition is selective for radial cells in lamina II and likely contributes to neuropathic pain. We suggest that a comparable circuit involving the CGRP radial cells could uncover low threshold inputs to ventrally located nociceptive circuits, which in recent years have been largely ignored (*Wercberger and Basbaum, 2019*).

# Materials and methods

## Animals

Mice were housed in cages on a standard 12:12 hr light/dark cycle with food and water ad libitum. Permission for all animal experiments was obtained and overseen by the Institutional Animal Care and Use Committee (IACUC) at the University of California San Francisco. All experiments were carried out in accordance with the National Institutes of Health Guide for the Care and Use of Laboratory Animals and the recommendations of the International Association for the Study of Pain.

## Mouse strains

The *Calca*$^{CreER}$ mouse strain was kindly provided by Dr. Pao-Tien Chuang (UC San Francisco) (*Song et al., 2012*). *Calca*$^{CreER}$ mice were then bred with C57BL/6J -Ai14 mice (Jackson Laboratory, Stock No: 007914) or with mice that selectively express green fluorescent protein (GFP) in gastrin-releasing peptide (GRP)-expressing cells (*Grp*$^{GFP}$ mouse *Solorzano et al., 2015*). *Lbx1*$^{FlpO}$ mice, in which FLPo is driven from the *Lbx1* promoter, were a kind gift from Dr. Martin Goulding at the Salk Institute, La Jolla CA.

## Tamoxifen

We dissolved tamoxifen (T5648, Sigma-Aldrich) in corn oil and injected it (150 mg/kg, i.p.) into the CGRP-tdTomato mice on two consecutive days. For immunohistochemistry, electrophysiology and tracing experiments we injected the tamoxifen into P21-22 mice. We waited 5 and 7–10 days before recording and perfusion for immunostaining, respectively. For Fluorogold (1%) tracing experiments, we injected the tracer into 6- to 8-week-old mice. For intraspinal surgeries intended for DREADD receptor expression studies, we injected tamoxifen into P11-12 mice and subsequently, between P14 and P16, made an intraspinal injection of hM3Dq without laminectomy.

## Fluorescence immunohistochemistry (IHC)

Mice of either sex were transcardially perfused with 10 mL phosphate-buffered saline (PBS) followed by 30 mL cold 4% formaldehyde in PBS. After dissection, dorsal root ganglia (DRG), trigeminal ganglia (TG), spinal cord, and caudal medullary tissue were post fixed for ~3 hr at room temperature and subsequently cryoprotected in 30% sucrose in PBS overnight at 4°C. The spinal cord and caudal medulla were sectioned in a cryostat at 25 μm; DRG and TG at 16 μm. After mounting and drying on slides, the sections were incubated for 1.5 hr in 10% normal goat serum with 0.3% Triton X-100 (NGST) to block non-specific antibody binding, and then for 24 hr in primary antibodies diluted in 10% NGST. The sections were then washed three times for 10 min in PBS and then incubated for 2 hr with a secondary antibody diluted in 1% NGST. After washing with PBS three times for 10 min, the sections were dried and coverslipped with Fluoromount G.

The following primary antibodies were used: rabbit anti-CGRP (1:1000, Peninsula), rabbit anti-calbindin (1:2000, Swant), mouse anti-calretinin (1:5000, Swant), guinea pig anti-PKCγ (1:7000, Strategic Bio), chicken anti-GFP (1:2500, Abcam), rabbit anti-Fos (1:5000, Calbiochem; 1:2000, Cell Signaling), guinea pig anti-Fluorogold (1:1000, Protos Biotech), guinea pig anti-Lmx1b (1:10000, kind gift from T. Müller and C. Birchmeier, Max Delbrück Center for Molecular Medicine, Berlin, Germany), rabbit anti-Pax2 (1:4000, Abcam), or rabbit anti HA (1:800, Cell Signaling). Secondary antibodies were conjugated to Alexa-488 or Alexa-647 (1:1000, Thermo Fisher Scientific).

## Capsaicin treatment

To ablate the central terminals of CGRP-expressing DRG neurons, CGRP-tdTomato mice were anesthetized with 2% isoflurane and injected intrathecally with 5.0 µl of a solution containing 10 µg of capsaicin, dissolved in 10% ethanol, 10% Tween-80% and 80% saline. Five days later, the mice received 5 i.p. injections of 150 mg/kg tamoxifen (one injection per day, on 5 consecutive days). Seven days later, the mice were processed for immunohistochemistry.

## Peroxidase immunocytochemistry

Mice were perfused with phosphate-buffered 4% formaldehyde (n = 3) or 4% formaldehyde plus 0.3% glutaraldehyde (n = 5). Transverse or parasagittal Vibratome sections (50 µm) were processed for detection of tdTomato for either light (LM) or electron microscopic (EM) (*Llewellyn-Smith et al., 2018*) examination.

## Electron microscopy

For EM analysis, the sections were washed for 2 hr in 50% ethanol, incubated for 30 min in 10% normal horse serum diluted with Tris-PBS (TPBS), then in 1:25,000 or 1:100,000 rabbit anti DSRed (Takara Bio USA) in 10% NHS-TPBS. The sections were subsequently exposed to 1:500 biotinylated donkey anti-rabbit IgG (Jackson ImmunoResearch) in 1% NHS-TPBS and then to 1:1500 ExtrAvidin-horseradish peroxidase (Sigma-Aldrich) in TPBS. Incubations in immunoreagents were for 3 days at room temperature on a shaker; sections were washed 3 × 30 min between incubations. To visualize CGRP-tdTomato-immunoreactivity in the dorsal horn, we used a nickel-intensified diaminobenzidine (DAB) reaction and hydrogen peroxide generated by glucose oxidase (*Llewellyn-Smith et al., 2005*). After the peroxidase reaction, sections containing tdTomato-immunoreactive neurons were osmicated, stained en bloc with aqueous uranyl acetate, dehydrated with acetone and propylene oxide, and infiltrated with Durcupan resin (Sigma-Aldrich). Finally, sections were embedded on glass slides under Aclar coverslips (Electron Microscopy Sciences) and polymerized at 60℃ for at least 48 hr. Dorsal horn regions containing CGRP-tdTomato neurons were re-embedded in resin on blank blocks under glass coverslips and repolymerized. Ultrathin sections were collected on copper mesh grids, stained with aqueous uranyl acetate, and examined with a JEOL 100CXII transmission electron microscope.

## LM analysis

Transverse or parasagittal Vibratome sections of tissue from mice perfused with phosphate-buffered 4% formaldehyde (n = 3) or 4% formaldehyde, 0.3% glutaraldehyde (n = 3) were either single stained to show tdTomato-immunoreactivity or double stained to demonstrate the relationships between VGLUT1-immunoreactive axons and CGRP-tdTomato neurons. All sections were washed 3 × 20 min in TPBS containing 0.3% Triton X-100 and exposed to 10% NHS in TPBS-Triton for 30 min. Single labeling involved exposure of sections to 1:25,000 or 1:100,000 anti-DSRed (Takara), 1:500 anti-rabbit IgG, 1:1500 ExtrAvidin-HRP and a nickel-intensified DAB reaction. For double labeling, VGLUT1-immunoreactivity was first detected with 1:50,000 or 1:100,000 rabbit anti-VGLUT1 (Synaptic Systems), biotinylated donkey anti-rabbit IgG, ExtrAvidin-horseradish peroxidase and a cobalt +nickel intensified DAB reaction (*Llewellyn-Smith et al., 2005*). Then, after another blocking step in 10% NHS, DSRed-immunoreactivity was detected as for single labeling except that the peroxidase reaction was intensified with imidazole (*Llewellyn-Smith et al., 2005*) rather than nickel. For LM labeling, primary antibodies were diluted with 10% NHS in TPBS-Triton; secondary antibodies, in 1% NHS-TPBS-Triton; and avidin-HRP complex, in TPBS-Triton. For LM, all incubations in immunoreagents were done on a shaker at room temperature for at least 24 hr and washes between incubations were 3 × 20 min in TPBS. Stained sections were mounted on subbed slides, dehydrated and coverslipped with Permaslip Mounting Medium (Alban Scientific).

## In situ hybridization (ISH)

In situ hybridization was performed using fresh spinal cord or caudal medullary tissue from adult mice (8–10 week-old), except for transient VGLUT3 assessment (*Peirs et al., 2015*), where the mice were 7 days old. We followed the protocol outlined by Advanced Cell Diagnostics (Newark, CA). The tissue was dissected out, instantaneously frozen on dry ice, and kept at –80℃ until use. Cryostat

sections of DRG (12 µm) were fixed at 4°C in 4% formaldehyde for 15 min, washed twice in PBS, and dehydrated through successive 5 min ethanol steps (50%, 70%, and 100%) and then dried at room temperature. After a 30 min incubation with protease IV, sections were washed twice in PBS and incubated at 40°C with RNAscope-labeled mouse probes: calcitonin gene-related peptide (*Calca*), RAR-related orphan receptor alpha (RORα), cholecystokinin (*Cck*), vesicular glutamate transporter 3 (*Slc17a8*), neurokinin receptor 1 (*Tacr1*), gastrin releasing peptide (*Grp*) for 2 hr in a humidified chamber. Sections were then washed twice in washing buffer and incubated with four 15–30 min 'signal amplifying' solutions at 40°C. After two washes, the sections were dried and covered with mounting media containing 4′,6-diamidino-2-phenylindole (DAPI).

## Image analysis

Images of fluorescent immunostained sections were acquired on an LSM 700 confocal microscope using ZEN Software (Carl Zeiss). The microscope was equipped with 405, 488, 555, and 639 nm diode lasers. For co-localization studies we used a 20x Plan-Apochromat (20×/0.8) objective (Zeiss) and image dimensions of 1024 × 1024 pixels with an image depth of 12 bits. Two times averaging was applied during image acquisition. Laser power and gain were adjusted to avoid saturation of single pixels and kept constant for each experiment. Image acquisition was performed with fixed exposure times for each channel and a 10% overlap of neighboring images where tiling was used. Stitching was done in ZEN using the 'stitching/fuse tiles' function. Adjustment of brightness/contrast and maximum projections of Z-stack images were done in Fiji/Image J. All images of the same experiment were processed in an identical manner.

Images of peroxidase immunostained sections were acquired on an Olympus BH2 brightfield microscope equipped with SPlanApo lenses and a SPOT Insight CMOS Color Mosaic 5MP camera running SPOT 5.3 Advanced software. For assessment of VGLUT1 appositions on DSRed-immunoreactive CGRP neurons, an x100 oil immersion lens was used. A VGLUT1-positive terminal was classified as forming a close apposition when (1) there was no space between the terminal and the DSRed-positive neuron for terminals lying side-by-side with a cell body or dendrite or when (2) the terminal and the DSRed-positive neuron were in the same focal plane for terminals overlying cell bodies or dendrites.

## Cell counts

To analyze overlap by immunohistochemistry or in situ hybridization, we counted cells from four to five sections in at least three animals per experiment. By immunohistochemistry, we first counted the number of neurons in the DRG and TG that were tdTomato-positive (total 1266 cells, three mice) or CGRP-positive (total 1050 cells, three mice) and then determined the percentage of tdTomato-positive neurons that were CGRP double-labeled and vice versa. The number of dorsal horn tdTomato-positive cells that double-labeled for different markers (e.g. PKCγ, Lmx1b, Fos, calretinin, calbindin) are indicated in the Results. To conclude that cells were double-labeled by in situ hybridization we set a threshold of at least five fluorescent 'dots' for each probe in conjunction with a DAPI-positive nucleus. Quantification of caspase-mediated ablation of CGRP-positive spinal cord neurons was performed in 5 Caspase-injected CGRP-tdTomato mice and four saline-injected, CGRP-tdTomato control mice. We counted neurons positive for tdTomato or PKCγ ipsilateral and contralateral to the injection side, in 5–10 sections per mouse, and then determined the percentage of tdTomato- or PKCγ-positive neurons in the ipsilateral side relative to the contralateral side.

## Viral vectors

For DREADD experiments we used a Cre and FlpO-dependent hM3D(Gq) adeno-associated virus: AAV1-hEF1alpha/hTLV1-Fon/Con[dFRT-HA_hM3D(Gq)-dlox-hM3D(Gq)-I-dlox-I-HA_hM3D (Gq)(rev)-dFRT]-WPRE-hGHp custom made by the University of Zurich Viral Vector Facility of the Neuroscience Center. For control injections, we used an AAV1.hSyn.eGFP.WPRE.SV40 from Addgene. For GCaMP-tracing experiments, we used an AV1.Syn.Flex.GCaMP6s.WPRE.SV40 from the Penn Vector Core, University of Pennsylvania. Note that we evaluated several Cre-dependent viral vectors for the tracing studies and only used those where specificity of expression was confirmed by lack of expression after injection into wild type mice. We waited at least 4 weeks to achieve stable viral expression before beginning the behavioral or neuroanatomical experiments. For Caspase-

mediated ablation experiments, we used a Cre-dependent adeno-associated virus-expressing Caspase (AAV1-flex-taCasp3-TEVp, titer: 1.5–2.8 × 10$^{12}$ viral particles/ml; Gene Therapy Vector Core at the University of North Carolina at Chapel Hill).

### Retrograde tracing

To study potential projection targets of the dorsal horn CGRP interneurons, we injected Fluorogold (1%) into several supraspinal sites known to receive projections from the spinal and medullary dorsal horns. We studied two mice for each location and allowed 5–9 days for tracer transport after which the mice were perfused with formaldehyde for subsequent histological analysis. We injected tracer into the following locations: ventrolateral thalamus (X:ML = 1.5, Y:AP = −1.82, Z:DV = 3.5; 500 or 800 nl); parabrachial nucleus (X = 1.25, Y = −4.95, Z = −3.6; 600 nl, see *Figure 5—figure supplement 2*); nucleus submedius of the thalamus (X = 0.5, Y = −1.43, Z = 4.25; 250 or 450 nl): dorsal column nuclei (400 nl).

### AAV injections

For all surgeries, the mice were administered carprofen (0.1 mg/kg, i.p.) just prior to surgery and lidocaine (0.5%) was applied to the incision site. For the DREADD experiments, under 2% isoflurane anesthesia, we injected P14-16 *Calca$^{CreER}$*-Lbx$^{FLPo}$ mice and littermates with an AAV-GFP. We removed muscles that overlay the left side of the T13 and L1 vertebra to expose the lumbar enlargement. Without laminectomy, we then slowly inserted a glass micropipette (50 μm tip) through the dura and made two 400 nl rostrocaudally separated injections of viral solution. The micropipette was left in place for ~2 min after which overlying muscle and skin were closed. After recovering from the anesthesia, the mice were returned to their home cages.

For the GCaMP6-tracing studies, we made injections (300–800 nl) into the medullary dorsal horn in 8-week-old mice anesthetized with i.p. ketamine (100 mg/kg) and xylazine (10 mg/kg) or isofluorane (2%). For injections into the nucleus caudalis, we incised the dura overlying the cisterna magna, exposing the caudal medulla and made a unilateral injection of viral solution with a glass micropipette. After recovering from anesthesia, the mice were returned to their home cage. For Caspase-mediated ablation studies, each mouse received a total of 2.0 μl of viral stock solution of the AAV1-flex-taCasp3-TEVp (ablated group) into the lumbar spinal cord. To measure post-virus mechanical and thermal thresholds, we tested the mice 3 weeks after virus injection, before and 7 and 14 days after spared nerve injury. At the end of all behavioral testing, control and ablated mice were euthanized, perfused and tissues harvested for quantification of the caspase-mediated ablation.

### Behavioral analyses

We took several measures to blind the behavioral experiments. (1) DREADD-injected and control (GFP-injected) mice were housed together. (2) A different experimenter performed the injections of CNO (5.0 mg/kg in saline) or saline before behavioral testing. (3) The behavioral tester recorded each mouse's eartag number after the test and was blind to the treatment (saline or CNO) that the mouse received or to which group the mouse belonged (AAV-GFP-injected or DREADD-injected). (4) Identification was made using records of eartag numbers after all testing was finalized.

### Static mechanical allodynia

For these experiments, we determined hindpaw mechanical thresholds with von Frey filaments, and quantified results using the updown method (*Chaplan et al., 1994*). The animals were habituated on a wire mesh for 2 hr on 2 consecutive days. On the next 2 days we recorded baseline thresholds, after a 1.5 hr of acclimatization on the wire mesh. After baseline determinations, the mice were injected with CNO or saline and then tested 30 min later. For all behavioral tests, either CNO or saline was injected every other day in a randomized fashion.

### Acetone test (cold allodynia)

Mice were habituated for 30 min on a mesh in plexiglass cylinders. Next, we used a syringe to squirt 50 μl acetone onto the plantar surface of the paw. The responses of the mice directly after application of acetone were recorded on video for 30 s. Each paw was tested five times and we measured

time (in seconds) spent lifting, licking or flinching the paw. Results are displayed as the average time across the five trials. Testing began 1 hr post injection of CNO or saline, with test days 48 hr apart.

## Hargreaves test

For thermal threshold testing (heat), we first acclimatized the mice for 30 min in Plexiglass cylinders. The mice were then placed on the glass of a Hargreaves apparatus and the latency to withdraw the paw from the heat source was recorded. Each paw was tested five times and we averaged latencies over the five trials. Hargreaves tests were done 1 hr after the tests of static dynamic mechanical allodynia.

## Pruritoception

We made a subcutaneous injection of 100 µl chloroquine (100 µg diluted in saline; Sigma-Aldrich) into the left calf. Mice were immediately placed into cylinders and video recorded for 30 min. Behavior was scored as number of scratching/biting bouts of the injection area over the 30 min.

## Rotarod test

Mice were acclimatized to the testing room and trained by placing them on an accelerating rotarod for a maximum of 60 s at low speed, three times with training taking place on two consecutive days. On testing days (48 hr apart), mice were injected with CNO or saline 30 min before being placed on the rotarod. Latency to fall was measured for up to 300 s. The procedure was repeated three times and latencies averaged across trials.

## Spared nerve injury (SNI)

To induce mechanical hypersensitivity in a model of neuropathic pain we used the approach described by *Shields et al., 2003*. Under isofluorane anesthesia (2%), two of the three branches of the sciatic nerve were ligated and transected distally, sparing the tibial nerve.

## Fos expression: capsaicin and formalin

To study the effects of a chemical algogen, we injected 10 µl of 2% formalin in saline into the cheek (n = 3). In a separate group of anesthetized animals (n = 3), we made a unilateral injection of 20 µl capsaicin (1.0 µg/µl) into the hindpaw or the cheek. We perfused all mice ~ 1.5 hr after injection and immunostained sections of the lumbar cord (paw injections) or caudal medulla (cheek injections) for Fos.

## Fos expression: chloroquine

To study the effects of a pruritogen, under isofluorane anesthesia, mice (n = 3) received unilateral injections of chloroquine (200 µg) into either the hindpaw (20 µl) or cheek (50 µl). The mice were perfused ~1.5 hr after injection and sections of the lumbar cord (paw injections) or caudal medulla (cheek injections) were immunostained for Fos.

## Fos expression: nitroglycerin

We injected mice (n = 3) with nitroglycerin (10 mg/kg, i.p.), which in humans can trigger a migraine and in rodents provokes behavioral signs of widespread thermal hyperalgesia and mechanical hypersensitivity (*Bates et al., 2010*), beginning 30–60 min after injection and subsiding within 4 hr. Based on this time course, the mice were perfused 2 hr after nitroglycerin injection and sections of caudal medulla were immunostained for Fos.

## Fos expression: dynamic mechanical allodynia

To assess Fos expression in uninjured animals (n = 3), we first acclimatized the mice to brushing of the cheek, (Utrecht 225, pure red sable brush 6, Germany) while lightly restraining the mouse in a towel with its head exposed. We brushed the left cheek along the direction of the hairs for 45 min, with a one minute break every 10 min. To monitor Fos expression in the injured animals, we performed unilateral partial sciatic nerve injury (SNI, see above). One week after SNI, we used a paintbrush (5/0, Princeton Art and Brush Co.) to lightly stroke the injured hind paw, from heel to toe (velocity:~2 cm/s). Ninety minutes to 2 hr after brushing, the mice were anesthetized, perfused and

spinal cord sections were immunostained for Fos. In a separate experiment, we also assessed Fos expression 1 week after SNI without applying a stimulus.

## Fos expression: rotarod test

Three mice were trained on a rotating rod for 60 min at a constant speed of 10 rpm. One week later the mice walked on the rotarod at 10 rpm for 1.5 hr (*Neumann et al., 2008*), after which they were anesthetized, perfused and lumbar spinal cord sections immunostained for Fos.

## Electrophysiology

Following our previous protocol (*Etlin et al., 2016*), we collected transverse lumbar and caudal medullary Vibratome (Leica) slices (350–400 µm) from 3 to 10 weeks old CGRP-tdTomato mice 5–7 days after tamoxifen injection. The sections were incubated in recording solution at 37°C for 1 hr and then transferred to a recording chamber (Automate Scientific) under an upright fluorescence microscope (E600FN; Nikon). The sections were superfused with recording solution at a rate of 1.0 ml/min and viewed with a CCD digital camera (Hamamatsu or DAGE-MTI). The transparent appearance of lamina II of the superficial dorsal horn and tdTomato-positive CGRP cells were obvious under near-infrared (IR) illumination. The patch pipettes were pulled to yield an impedance of 6–8 MΩ on a horizontal pipette puller (Sutter Instrument) from thin-walled, fire-polished, borosilicate glass filaments. The pipette solution composition was (in mM): K-methane sulfonate 140, NaCl 10, CaCl$_2$ 1.0, EGTA 1.0, HEPES 10, Mg-ATP 5.0, and NaGTP 0.5 and included 5.0 mg/ml of Biocytin (Sigma-Aldrich) for intracellular filling of the recorded cells. Neurons were approached with a micromanipulator (Sutter Instrument) while monitoring the resistance in voltage-clamp mode using the 'Membrane Test' module of pClamp10 software (Molecular Devices). To prevent clogging of the tip, we applied positive pressure to the pipette via a 1.0 ml syringe. After a seal was established with a cell, we ruptured its membrane by gently applying negative pressure to the pipette to secure a whole-cell configuration. Current and voltage signals were amplified using a DC amplifier (MultiClamp 700) and digitized using Digidata 1440a system (Molecular Devices) at 10 kHz and then stored for subsequent offline analysis.

In some experiments, we placed an attached dorsal root in a suction electrode to be stimulated electrically while simultaneously measuring evoked responses of the tdTomato-expressing neurons. To determine the fiber types providing input to the recorded neurons, and to assess the monosynaptic/polysynaptic nature of the Aβ, Aδ, and C fiber inputs, the dorsal roots were stimulated 20 times at the following frequencies and intensities (25 µA, 20 Hz for Aβ fibers; 100 µA, 2 Hz and occasionally 10 Hz for Aδ fibers; 500 µA, 1 Hz for C fibers).

For current clamp recordings, after whole cell configuration was achieved, action potentials were induced by current steps, from −10 to 150 pA, with an increment of 5.0 or 10 pA (pulse duration 300 ms). The rheobase was determined using a 5.0 pA increment current step (pulse duration 300 ms). Only neurons with a resting membrane potential of at least −40 mV and stable baseline were used for further experiments and analysis. The recording was abandoned with loss of spike overshoot. To determine the effect of the inhibitory inputs on excitability of the CGRP interneurons, slices were continuously perfused with 20 µM bicuculline and 4.0 µM strychnine and the rheobase measured before and after application of the GABA and glycine antagonists. After establishing a stable baseline recording, we maintained the neurons 10 pA below rheobase (pulse duration 300 ms, sweep intervals of 30 s), a current at which action potentials were never evoked. Next the bicuculline/strychnine solution was applied to the recording chamber. The appearance of an action potential signaled that the antagonists had removed a tonic inhibition of the CGRP interneurons.

## Statistical analysis

Statistical analyses were performed using SPSS (IBM-SPSS version 24). Similarity of normality and variance were assessed before applying parametric or non-parametric tests. For analysis of the effect of CNO on mechanical hypersensitivity, we assessed interaction between treatment (CNO, saline or baseline) with group (DREADD-virus injected animals or GFP-virus injected animals) by repeated measures two-way ANOVA, including all conditions and groups. Statistics were calculated based on a type III sum of squares model and significant interaction effects were assessed using deviation from the mean of the control groups. The N was estimated based on variance for von Frey

experiments using an a priori power calculation. Hargreaves and rotarod results were analyzed using Student's t-tests. For acetone sensitivity we used the Wilcoxon signed rank test. Parametric and non-parametric tests are reported as mean ± SEM or by medians and inter-quartiles, respectively. Electrophysiological recordings of intrinsic membrane and action potential properties were calculated using custom-written Matlab scripts (MathWorks, Illinois) as previously described (*Etlin et al., 2016*). p Values were considered significant if $p < 0.05$.

## Acknowledgements

This research was supported by a Sir Henry Wellcome Fellowship 092208/Z/10/Z (LSL), Åke Wiberg Foundation (LSL), NIH: R35NS097306 (AIB), Wellcome Award: A102645 (AIB) and Open Philanthropy (AIB). We are grateful to Dr. Hendrik Wildner, University of Zurich for sharing the Cre/Flp dependent DREADD construct and to Dr. Ling Bai, University of California San Francisco for helpful advice on surgeries.

## Additional information

### Competing interests

Allan Basbaum: Reviewing editor, *eLife*. The other authors declare that no competing interests exist.

### Funding

| Funder | Grant reference number | Author |
| --- | --- | --- |
| Wellcome Trust | 092208/Z/10/Z | Line S Löken |
| Åke Wiberg Foundation | | Line S Löken |
| NIH Blueprint for Neuroscience Research | R35NS097306 | Allan Basbaum |
| Wellcome Trust | A102645 | Allan Basbaum |
| Open Philanthropy Project | | Allan Basbaum |

The funders had no role in study design, data collection and interpretation, or the decision to submit the work for publication.

### Author contributions

Line S Löken, conceptualized and designed the study, performed the experiments, collected the data, analyzed data and wrote the manuscript; Joao M Braz, conceptualized and designed the study, performed experiments and collected data, analyzed data and wrote the manuscript; Alexander Etlin, Ida J Llewellyn-Smith, performed the experiments, collected the data, analyzed data and wrote the manuscript; Mahsa Sadeghi, performed experiments, collected and analysed the data; Mollie Bernstein, Madison Jewell, Marilyn Steyert, Julia Kuhn, Katherine Hamel, performed the experiments and collected the data; Allan Basbaum, conceptualized and designed the study, analyzed data and wrote the manuscript

### Author ORCIDs

Line S Löken  https://orcid.org/0000-0001-6762-9717
Mahsa Sadeghi  https://orcid.org/0000-0002-9769-4082
Ida J Llewellyn-Smith  http://orcid.org/0000-0003-4269-6846
Allan Basbaum  https://orcid.org/0000-0002-1710-6333

### Ethics

Animal experimentation: Mice were housed in cages on a standard 12:12 hour light/dark cycle with food and water ad libitum. Permission for all animal experiments was obtained and overseen by the Institutional Animal Care and Use Committee (IACUC) at the University of California San Francisco. All experiments were carried out in accordance with the National Institutes of Health Guide for the

Care and Use of Laboratory Animals and the recommendations of the International Association for the Study of Pain. Ethical approval number: AN183265; expires Feb. 26, 2023.

## Decision letter and Author response

Decision letter https://doi.org/10.7554/eLife.59751.sa1
Author response https://doi.org/10.7554/eLife.59751.sa2

## Additional files

### Supplementary files

• Transparent reporting form

### Data availability

All data generated or analysed during this study are included in the manuscript and supporting files. Source data file is available for figure 6.

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
