## [Decision Letter]

**Acceptance summary:**

This study uncovers an interesting and important population of spinal dorsal horn neurons expressing calcitonin gene releasing peptide (CGRP), and describes a novel source for this important nociceptive peptide. Using a variety of excellent methodologies, including viral tracing, electrophysiology and chemogenetics, the study describes a novel facet of spinal circuitry mediating mechanical pain and allodynia in neuropathic conditions.

**Decision letter after peer review:**

Thank you for submitting your article "Dorsal horn CGRP-expressing interneurons contribute to nerve injury-induced mechanical hypersensitivity" for consideration by *eLife*. Your article has been reviewed by 3 peer reviewers, one of whom is a member of our Board of Reviewing Editors, and the evaluation has been overseen by Catherine Dulac as the Senior Editor. The reviewers have opted to remain anonymous.

The reviewers have discussed the reviews with one another and the Reviewing Editor has drafted this decision to help you prepare a revised submission.

Summary:

The manuscript by Löken et al. describes a novel type of interneurons expressing calcitonin gene releasing peptide (CGRP) in the spinal dorsal horn. The study characterizes a new group of dorsal horn neurons that may mediate nerve injury-induced mechanical hypersensitivity. These findings hold promise in understanding spinal mechanisms of pain and allodynia. The utilizes a number of state-of-the-art methodologies, including viral tracing, electrophysiology and chemogenetics.

The reviewers have reached agreement that the study has potential for *eLife*, but is premature in its current form. Given the restrictions imposed by the Covid crisis, the reviewers understand that performing major revisions may not be feasible and have therefore endeavored to find a minimal common basis for new experiments that would allay most of the major concerns.

Essential revisions:

The following changes are required:

1. There is consensus that silencing, rather than activating this subtype of CGRP cells, would strongly help understanding the role of these CGRP spinal neurons. This will require new experiments, but it seems essential to support the main conclusions of this study. This experiment would also help address other concerns the reviewers had.

2. There were multiple concerns regarding the electrophysiology experiments and c-Fos analyses shown in the manuscript. The reviewers have reached an agreement that most of these could be addressed by performing additional slice recordings intracellular recordings in current clamp mode to determine if the Aβ inputs to CGRP neurons are gated via feed forward inhibition and fail to produce action potential firing. These would also show subthreshold activity engaged by low threshold electric stimulation in physiological conditions, but not after injury and help explain the lack of c-Fos induction by sustained or repeated low threshold mechanical stimuli under naive conditions.

3. Missing controls (e.g. sense controls in mRNA in situ hybridization and c-Fos expression) must be addressed and low sample size in electrophysiology experiments should be discussed and appropriately acknowledged in the discussion. An N of 3 mice for controls in behavioral experiments is not adequate.

4. Thorough changes are required in the text to correct misleading generalizations, e.g. evidence for predominant Aβ input is too weak given the low n number (n = 5 cells) as well as the observation that 40% of these did respond to Aβ inputs.

5. There are concerns about the lack of CGRP immunoreactivity in td-Tomato-positive putative CGRP neurons. The reviewers understand that additional analyses (e.g. CGRP-immunoreactivity in terminals of td-Tomato neurons or EM analyses) may be cumbersome, but this discrepancy should be at least acknowledged and discussed thoroughly in a balanced manner in the manuscript.

---

## [Author Response]

Essential revisions:The following changes are required:1. There is consensus that silencing, rather than activating this subtype of CGRP cells, would strongly help understanding the role of these CGRP spinal neurons. This will require new experiments, but it seems essential to support the main conclusions of this study. This experiment would also help address other concerns the reviewers had.

We agree with the reviewers and now provide results of an extensive set of behavioral and anatomical experiments performed in animals in which we used a caspase-mediated approach to selectively ablate the CGRP interneurons from the dorsal horn of the spinal cord. Consistent with what we hypothesized in the original manuscript, ablation of the CGRP interneurons selectively affected baseline mechanical responses to von Frey filaments. Responses to noxious thermal (heat and acetone) and pruritic stimuli did not differ. These findings corroborate our previous conclusion that the CGRP-expressing interneurons constitute a separate class of excitatory interneurons that are mechanically driven and contribute to the reflex circuits in the normal mouse.

2. There were multiple concerns regarding the electrophysiology experiments and c-Fos analyses shown in the manuscript. The reviewers have reached an agreement that most of these could be addressed by performing additional slice recordings intracellular recordings in current clamp mode to determine if the Aβ inputs to CGRP neurons are gated via feed forward inhibition and fail to produce action potential firing. These would also show subthreshold activity engaged by low threshold electric stimulation in physiological conditions, but not after injury and help explain the lack of c-Fos induction by sustained or repeated low threshold mechanical stimuli under naive conditions.

We appreciate the importance of determining whether there is a tonic inhibitory regulation of the CGRP interneurons. To this end, we have significantly increased the number of neurons from which recordings were made, and most importantly, evaluated the effects of blocking any baseline inhibitory controls by incubating the slice with antagonists of GABAergic and glycinergic receptors. We found that under normal conditions, the CGRP interneurons receive a tonic inhibitory input. This was demonstrated in current clamp mode, in which we document that there is both a significant reduction in rheobase as well as an increase in resting membrane potential in the presence of the antagonists.

3. Missing controls (e.g. sense controls in mRNA in situ hybridization and c-Fos expression) must be addressed and low sample size in electrophysiology experiments should be discussed and appropriately acknowledged in the discussion. An N of 3 mice for controls in behavioral experiments is not adequate.

As described in the Methods section, we performed in situ hybridization according to ACD’s protocol using RNAscope probes that are designed to ensure specificity and sensitivity. The specificity is achieved by the double Z probe strategy that requires a probe pair (ZZ) to simultaneously bind to the target in order to generate signal. Non-specific binding of single Z probes will not yield detectable signals. Since all RNAscope probes are designed by applying the same screening criteria and the assays are run under the same stringent conditions (hybridization and washing), a similar level of high specificity is expected for all probes. In addition, during the assay, to assess any nonspecific background in a particular run/sample, a negative control probe (bacterial gene dapB) was run in parallel with the same sample and yielded no signal. ACD’s website is an excellent source for more detailed information relating to probe design, selectivity and sensitivity: https://acdbio.com/science/how-it-works.

Concerning sample size: as noted above we have significantly increased the number of CGRP positive interneurons from which recordings were made. In addition, with respect to the number of mice in behavioral experiments, please note that there were equal numbers of male and female mice (3 each). Thus, controls, in fact, contained 6 mice. And in the caspase studies included in the revision, sample sizes were 8 ablated and 6 controls.

4. Thorough changes are required in the text to correct misleading generalizations, e.g. evidence for predominant Aβ input is too weak given the low n number (n = 5 cells) as well as the observation that 40% of these did respond to Aβ inputs.

We appreciate the concern as to number of neurons in what is a difficult experiment. In fact, the Results section and Figure 6 - source data 1 show that all 5 of the 5 neurons tested responded to Aβ inputs. In contrast, only 2 of these neurons could be activated by intensities that we presume engaged C fibers. We have modified the text to make these findings clearer.

5. There are concerns about the lack of CGRP immunoreactivity in td-Tomato-positive putative CGRP neurons. The reviewers understand that additional analyses (e.g. CGRP-immunoreactivity in terminals of td-Tomato neurons or EM analyses) may be cumbersome, but this discrepancy should be at least acknowledged and discussed thoroughly in a balanced manner in the manuscript.

We agree with the Reviewers that there is indeed a discrepancy between the CGRP mRNA and protein expression levels in the spinal cord. Although we were successful in detecting strong CGRP mRNA expression using *in situ* hybridization and strong Tdtomato expression in the CGRP-Cre-Tdtomato reporter mouse (in which the reporter gene is under the influence of the Calca gene promoter), to date our attempts to detect the CGRP peptide by immunohistochemistry have failed. However, as we wrote in the submitted manuscript the Hökfelt group did detect CGRP immunoreactivity in some dorsal horn spinal cord neurons (Tie-Jun et al., 2001). Their studies required colchicine treatment. As we wrote in the Results section: “the lack of CGRP immunostaining reflects rapid transport of the peptide from the cell body to its axon, which undoubtedly underlies the requirement for colchicine to demonstrate these neurons by immunocytochemistry”.